# Augmented Mixup Procedure for Privacy-Preserving Collaborative Training

**Mihail Plesa**                                    *mihail.plesa@orange.com*
*Orange Services*

**Fabrice Clérot**                                  *fabrice.clerot@orange.com*
*Orange Research*

**Simona David**                                    *simona1.david@orange.com*
*Orange Services*

**Robert Poenaru**                                  *robert.poenaru@orange.com*
*Orange Services*

**Reviewed on OpenReview:** *https://openreview.net/forum?id=1SrZyNgmpY*

## Abstract

Mixup involves training neural networks on convex combinations of input samples and labels and has been adapted for privacy-preserving collaborative training, most notably in InstaHide. However, mixing-based obfuscation schemes create structured linear systems that can be exploited to reconstruct the underlying private data. We propose a singularized mixup procedure that injects controlled perturbations prior to forming convex combinations, rendering the resulting inverse problem ill-conditioned while preserving discriminative structure. We provide an average-case theoretical analysis that characterizes the security–utility trade-off via minimax reconstruction bounds and directional signal-to-noise ratio control. Empirically, we evaluate classification accuracy on MNIST, CIFAR-10, CIFAR-100, and Tiny-ImageNet, and compare against InstaHide, observing competitive or improved accuracy under strong privacy settings. We assess robustness against both linear and nonlinear reconstruction attacks, including at-scale linear inversion experiments on CIFAR-5M. In a collaborative training setting with multiple parties and heterogeneous data partitions, we further compare against standard federated learning (FedProx), showing that singularized mixup enables accurate centralized training without iterative gradient exchange and yields improved robustness and performance in heterogeneous regimes. Overall, our results demonstrate that singularized mixup substantially degrades reconstruction quality while maintaining strong predictive performance, providing a practical and scalable approach to privacy-preserving collaborative learning.

## 1 Introduction

Data mixing was initially introduced as a dataset augmentation technique, generating new samples by computing weighted averages of subsets from the original dataset Zhang et al. (2017). Originally designed as a regularization method for training neural networks, this approach has also been adapted for privacy-preserving protocols, as the mixing process obscures the original data during model training Liu et al. (2019); Fu et al. (2019).

Although the mixup strategy appears to preserve privacy without significantly degrading model performance, directly applying the method proposed by Zhang et al. (2017) can introduce vulnerabilities that allow attackers to recover private data under certain conditions Huang et al. (2020):

1. **Mixup samples from a private dataset only:** If mixup samples are generated exclusively from a private dataset, an attacker can identify which samples share a common private component by analyzing the expected value of the dot product between mixup samples. Once a set of related mixup samples is identified, the common private sample can be reconstructed by averaging these samples.

2. **Mixup samples from both private and public datasets:** When mixup samples are generated using both private and public datasets, repetitions of private samples in the mixup process can be avoided by leveraging the public dataset. However, since the public dataset is accessible, an attacker can perform a similar statistical analysis to identify which public samples were used in the mixup. Once the public components are determined, the private sample can be trivially reconstructed.

Building on their security analysis, Huang et al. (2020) proposed a mixup-based algorithm called InstaHide. The key innovation of their method is the application of a sign-flipping mask to images generated by computing a weighted sum of both public and private samples. The authors analyzed the security of InstaHide and formally proved that its security depends on the computational hardness of the subset-sum problem. However, the assumptions underlying their security model do not accurately reflect the properties of real-world data. In particular, Huang et al. (2020) assumed that each sample consists of an arbitrary sequence of values. For example, in the context of images, this assumption implies that pixels are independently and randomly distributed, which does not hold in practice. Consequently, although the security proof is mathematically valid, it does not offer practical security guarantees. This limitation was highlighted by Carlini et al. (2021a), who developed efficient attacks on samples generated by the InstaHide algorithm, enabling near-complete recovery of the original data.

## 2 Contributions

The primary contribution of this paper, presented in Section 3, is a singularized mixup algorithm that resolves the core weakness exploited by major attacks on InstaHide: the repeated reuse of the same private sample across mixtures. Our method mixes only two private images at a time and injects structured noise into the non-target component, thereby eliminating the persistent signal required for reconstruction attacks.

In Section 4, we develop a theoretical security analysis of the proposed mechanism. We establish lower bounds on the achievable reconstruction error under an adversary with full knowledge of the mixing weights, and we provide a principled way to choose the noise norm so that the SNR associated with separating each encoded sample into signal and interference components remains below a prescribed security parameter $\tau$.

Section 5 presents a comprehensive empirical evaluation of both security and utility. We assess robustness against linear and nonlinear attackers under a conservative threat model and show that, above a modest noise threshold, neither can reliably recover the underlying private images. At the same time, even when using a noise level substantially stricter than required by the theoretical bounds, our method maintains accuracy comparable to InstaHide's strongest $k = 4$ configuration on MNIST, CIFAR-10, CIFAR-100, and Tiny-ImageNet. We further demonstrate in collaborative training experiments that singularized mixup consistently outperforms a federated baseline (FedProx) under varying degrees of label and size heterogeneity, while avoiding iterative gradient exchange. Overall, the results indicate that the proposed mechanism provides strong empirical protection against reconstruction attacks while preserving high downstream utility in both standalone and collaborative regimes.

## 3 Singularized mixup

In this section, we introduce our algorithm, which is based on the singularization framework. This approach enhances security by ensuring that each execution is unique, thereby making it broadly applicable to a variety of systems Gaber et al. (2023). Singularization has previously been used to strengthen encryption algorithms without modifying their underlying structure Macario-Rat & Plesa (2024), which motivates our adoption of this framework in the design of our mixup algorithm.

We begin with a brief overview of InstaHide, followed by a detailed description of our proposed algorithm.

## 3.1 InstaHide algorithm

Consider a private dataset $(x_i, y_i)_{i=1}^{n}$ consisting of $n$ samples, where $x_i \in \mathbb{R}^d$ denotes the input example and $y_i \in \mathbb{R}^c$ is the corresponding one-hot encoded label.

The fundamental idea behind Mixup, as introduced by Zhang et al. (2017), is to replace each data point with a convex combination of the current sample and $k-1$ other samples selected uniformly at random from the dataset. Specifically, each new data point is generated by taking a weighted average of $k$ instances and their associated labels:

$$\tilde{x}_i \leftarrow w_{i1}x_i + \sum_{j=2}^{k-1} w_{ij}x_{\pi_i(j)} \tag{1}$$

$$\tilde{y}_i \leftarrow w_{i1}y_i + \sum_{j=2}^{k-1} w_{ij}y_{\pi_i(j)} \tag{2}$$

where $\{(\tilde{x}_i, \tilde{y}_i)\}_{i=1}^{n}$ represents the encoded dataset and $\pi_i$ is a random permutation over $\{1, 2, \ldots, n\}$.

The InstaHide approach builds upon Mixup but introduces two key modifications:

1. **Public images**: InstaHide augments the private dataset with samples from public datasets, expanding the pool of mixing samples to $(x_i, y_i)_{i=1}^{n} \cup (x_i, y_i)_{i=n+1}^{n+m}$, where $m$ denotes the size of the public dataset.

2. **Sign mask**: The sign of each pixel in a mixup image is randomly flipped using a random sign mask $\sigma_i \sim \Lambda_{\pm}^{d}$.

As a result, equations (1) and (2) are modified as follows:

$$\tilde{x}_i \leftarrow \sigma_i \circ \left( w_{i1}x_i + \sum_{j=2}^{k_s-1} w_{ij}x_{\pi_i(j)} + \sum_{j=k_s+1}^{k-k_t} w_{ij}x_{\pi_{i_p}(j)} \right) \tag{3}$$

$$\tilde{y}_i \leftarrow w_{i1}y_i + \sum_{j=2}^{k_s-1} w_{ij}y_{\pi_i(j)} \tag{4}$$

Here, $k_s$ denotes the number of private images, $k_t$ the number of public images, and $\pi_{i_p}$ is a random permutation over the set $\{n+1, \ldots, n+m\}$. Note that public images are used solely as a source of structured noise, and their labels are not included in the mix.

## 3.2 Singularization algorithm

The principal aim of the singularization algorithm is to transform the original dataset $\{(x_i, y_i)\}_{i=1}^{n}$ into a new set $\{(\tilde{x}_i, \tilde{y}_i)\}_{i=1}^{n}$ such that the resulting dataset preserves the discriminative characteristics of the original data, while ensuring that the original data cannot be recovered.

A key vulnerability of the InstaHide algorithm arises from the possibility that two encoded samples may share the same private input during the mixup process. This stems from InstaHide's encoding strategy, where each encoded sample is formed as a convex combination of two data points from the original private dataset and $k-2$ additional samples drawn from public sources. The inclusion of multiple private samples in each encoded combination allows an attacker to group encoded samples that share a common private component, making it possible to reconstruct the original private image from these clusters. For the attacker, the

repeated appearance of a private sample across different encodings acts as a persistent signal, while the public components serve as noise that can be filtered out through aggregation.

There are three primary distinctions between our approach and InstaHide. First, our method constructs each encoded input using exactly two private data points ($k = 2$), and does not incorporate any public data. This design choice is motivated by security considerations. While InstaHide suggests increasing $k$ to mitigate brute-force attacks on the subset sum problem, Carlini et al. (2021a) has demonstrated that larger values of $k$ can actually reduce security. Specifically, when the mixing weights are known, the resulting linear system becomes easier to solve, making it more vulnerable to attack. By fixing $k = 2$ and relying exclusively on private data, our approach avoids these vulnerabilities.

Second, we introduce noise only to the second private data point before performing the mixup operation. This strategy is intended to tightly couple the noise with the private information, thereby making it significantly more difficult for an adversary to recover the original data. When the noise level is sufficiently high, inverting the process becomes an ill-conditioned problem, which further enhances security.

Finally, our method does not require sign-flipping masks or the use of public images. Previous attacks Carlini et al. (2021a); Chen et al. (2020); Luo et al. (2022) have shown that sign-flipping masks can be circumvented by analyzing the absolute values of mixed images, thus limiting their effectiveness as a privacy mechanism. Our singularization algorithm is presented in Algorithm 1.

---

**Algorithm 1** Singularized Mixup

---

**Require:** Dataset $\{(x_i, y_i)\}_{i=1}^n$; error norm $r$
**Ensure:** Mixed dataset $\{(\tilde{x}_i, \tilde{y}_i)\}_{i=1}^n$
1: $\pi \sim \mathrm{Uniform}(S_n)$
2: **for** each $i = 1$ to $n$ **do**
3:      $w_i \sim \mathrm{Uniform}([0,1]^2)$ and normalize such that $\|w_i\|_1 = 1$ and $\|w_i\|_\infty \leq \alpha$
4:      $e_i \sim \mathrm{Uniform}(\mathbb{S}(0, r))$
5:      $\tilde{x}_i \leftarrow w_{i1} x_i + w_{i2}(x_{\pi(i)} + e_i)$
6:      $\tilde{y}_i \leftarrow w_{i1} y_i + w_{i2} y_{\pi(i)}$
7: **end for**
       **return** $\{(\tilde{x}_i, \tilde{y}_i)\}_{i=1}^n$

---

### 3.3 Practical Instantiation

Similar to InstaHide Huang et al. (2020), the primary application of our algorithm is in privacy-preserving collaborative training. Suppose there are multiple parties, each possessing a private local dataset. These parties aim to jointly train a deep neural network on the combined data without exposing the sensitive information contained in their individual datasets. The following general framework demonstrates how Algorithm 1 can be integrated to achieve this goal:

1. All parties agree on a common preprocessing technique to be applied locally. For instance, in the context of image data, participants may choose to standardize the images or extract feature representations using a publicly available pretrained model, such as ResNet He et al. (2016).

2. Each party independently transforms its local dataset by applying Algorithm 1, thereby generating a set of mixup samples. Each sample consists of a mixup example and its corresponding mixup label.

3. The resulting data is then transmitted to a central server, which is responsible for training the deep learning model. Upon completion, the trained model is distributed back to the parties for local use.

The security of this protocol depends on the effectiveness of Algorithm 1 in protecting the privacy of local datasets. In particular, the central server, which only receives the mixup samples, should not be able to reconstruct the original data from these representations.

# 4 Security analysis

In this section, we analyze the security of the proposed scheme and evaluate its resilience to the three principal attacks on the InstaHide framework, as presented in Carlini et al. (2021a), Chen et al. (2020), and Luo et al. (2022). We begin by reviewing the attack strategy of Carlini et al. (2021a), which applies directly to InstaHide without additional assumptions and forms the foundation for subsequent attacks.

The attack of Carlini et al. (2021a) aims to recover the noisy linear system of equations generated by the InstaHide encoding procedure. The adversary's first task is to identify which private images contribute to each encoded sample. To do so, the attacker generates encoded images using publicly available data and trains a neural network to predict whether two encoded images share a common private source image. Although InstaHide introduces random sign flips that might be expected to impede this process, the authors show that taking the absolute value of each pixel effectively removes this obstacle. Once trained, the comparison network allows the adversary to infer the co-occurrence of private images across encoded samples. Since the mixing weights are revealed through the encoded labels, the attacker can then assemble a noisy linear system whose noise arises from the public images included in each mixup. Because the public images vary across samples, this noise behaves approximately as mean-zero Gaussian and averages out across many equations. Solving this system enables the attacker to reconstruct the private images up to a global sign per pixel. A final recoloring step is applied to improve visual fidelity.

In contrast, the attack of Chen et al. (2020) relies on an explicit distributional assumption: the original images are modeled as Gaussian. Under this assumption, the absolute values of the encoded samples follow a folded Gaussian distribution. Given sufficiently many mixup samples, the adversary can estimate their covariance matrix, which corresponds to the Gram matrix of the mixing weight vectors. From this Gram matrix, the adversary can determine which private images participate in each mixup. As in Carlini et al. (2021a), this identification step enables the construction of a linear inverse problem whose solution yields the private images up to pixel-wise sign ambiguities.

To mitigate the attack of Carlini et al. (2021a), Luo et al. (2022) proposes introducing geometric augmentations—such as shifting, cropping, rotation, and translation—to disrupt pixel-wise alignment before mixup. This defense seeks to prevent the formation of a consistent linear system that attackers could invert. Their approach shares a broad motivation with ours: both aim to ensure that identical images do not reappear across multiple encoded samples. However, Luo et al. (2022) demonstrates that such augmentations can be circumvented. Specifically, the attacker trains a comparison network, as in Carlini et al. (2021a), to detect whether two encoded samples share a common private image, even under geometric transformations. Encoded samples containing the same private image are then clustered. Within each cluster, a fusion-denoising pipeline is applied: a convolutional network first downsamples the encoded images to reduce geometric variability, a transpose CNN upsamples the result, and multiple outputs are fused (by averaging or max-pooling) before being passed through a denoising network that reconstructs the private image.

The attacks of Carlini et al. (2021a) and Chen et al. (2020) are ineffective against our method because the injected noise disrupts the formation of stable linear systems. Moreover, unlike Luo et al. (2022), our scheme does not rely on geometric transformations, thereby limiting the applicability of fusion-denoising attacks. Although our mechanism is designed to prevent adversaries from forming clusters or reconstructing underlying linear systems, we adopt a conservative security model in which the adversary is assumed capable of doing so. This assumption is motivated by a common feature of all three attacks: the ability to determine whether two encoded images share a private component. In both Carlini et al. (2021a) and Chen et al. (2020), the random sign mask is rendered ineffective by taking absolute values, while in Luo et al. (2022), geometric transformations do not prevent clustering.

To evaluate the security of our singularized mixup algorithm, we analyze the minimax reconstruction error encountered by an adversary attempting to invert the system of equations induced by Algorithm 1. Specifically, we consider an attacker with access to both the encoded samples and the mixing weights, and who seeks to recover the original data. Theorem 1 establishes a lower bound on the expected Euclidean recovery error that holds uniformly over all possible estimators, including both linear and non-linear strategies. This minimax bound is derived under the assumption that the adversary does not exploit any structural

prior information about the data, an assumption that is standard in the literature Huang et al. (2020). To complement this theoretical analysis and address scenarios where the attacker may possess such priors, we conduct extensive empirical evaluations using both linear and non-linear adversaries on image datasets.

The theorem demonstrates that the minimax expected recovery error scales linearly with the noise radius $r$, thereby establishing a formal obstruction to accurate reconstruction under an $\ell_2$ metric. We emphasize that this notion of security is not cryptographic: unlike definitions such as semantic security, which aim to prevent an adversary from learning *any* information about the underlying plaintext, our guarantee quantifies a lower bound on reconstruction accuracy. Achieving such cryptographic guarantees typically requires algorithms based on homomorphic encryption or secure multi-party computation, which incur substantial computational and communication overhead and severely limit scalability in collaborative learning settings. In contrast, our security metric captures the practical requirement that an attacker should not be able to recover *meaningful* information about the original samples, while preserving efficiency and enabling large-scale training without operating on encrypted data.

**Theorem 1.** *Let $X \in \mathbb{R}^{n \times d}$ have rows $x_i^\top$. Algorithm 1 produces*

$$\tilde{x}_i = w_{i1}\, x_i + w_{i2}\, (x_{\pi(i)} + e_i), \qquad e_i \overset{i.i.d.}{\sim} \text{Uniform}(\mathbb{S}(0, r)),$$

*with $\|w_i\|_1 = 1$ and $\|w_i\|_\infty \le \alpha$. Let $P_\pi$ be the permutation matrix of $\pi$ and define*

$$D_1 := \text{diag}(w_{11}, \dots, w_{n1}), \qquad D_2 := \text{diag}(w_{12}, \dots, w_{n2}), \qquad W := D_1 + D_2 P_\pi,$$

*and assume $W$ is invertible. Then*

$$\tilde{X} = WX + E, \qquad E_i = w_{i2}\, e_i.$$

*For any estimator $\hat{x}_i = \hat{x}_i(\tilde{X}, W)$,*

$$\sup_{X \in \mathbb{R}^{n \times d}} \mathbb{E}\big[\|x_i - \hat{x}_i(\tilde{X}, W)\|_2^2\big] \ge r^2\, T_i,$$

*where*

$$T_i := \sum_{\ell=1}^{n} (W^{-1})_{i\ell}^2\, w_{\ell 2}^2.$$

*Proof.* Since $W$ is invertible, define

$$Y := W^{-1}\tilde{X} = X + Z, \qquad Z := W^{-1}E.$$

Because $E_\ell = w_{\ell 2} e_\ell$ and $e_\ell$ are independent, mean-zero, and isotropic with

$$\text{Cov}(e_\ell) = \frac{r^2}{d} I_d,$$

we have

$$\text{Cov}(E_\ell) = w_{\ell 2}^2\, \frac{r^2}{d}\, I_d.$$

Thus

$$\text{Cov}(Z_i) = \sum_{\ell=1}^{n} (W^{-1})_{i\ell}^2\, \text{Cov}(E_\ell) = \frac{r^2}{d}\, T_i\, I_d,$$

and hence

$$\mathbb{E}\|Z_i\|_2^2 = \text{tr}(\text{Cov}(Z_i)) = r^2\, T_i.$$

Fix any estimator $\hat{x}_i(\tilde{X}, W)$ and write $\delta(Y) := \hat{x}_i(\tilde{X}, W)$. Consider matrices $X$ with all rows equal to zero except the $i$th. For such $X$,

$$Y_i = x_i + Z_i, \qquad (Y_{-i}, Z_i) \text{ independent of } x_i.$$

Let $m(Y_i) := \mathbb{E}[\delta(Y) \mid Y_i]$. By Jensen's inequality,

$$\mathbb{E}\big[\|x_i - m(Y_i)\|_2^2\big] \ \le \ \mathbb{E}\big[\|x_i - \delta(Y)\|_2^2\big] \qquad \text{for every } x_i \in \mathbb{R}^d.$$

Now restrict attention to this one-dimensional family $X$ parametrized by $x_i$. We obtain the $d$-dimensional location model

$$Y_i = x_i + Z_i, \qquad x_i \in \mathbb{R}^d,$$

with fixed noise $Z_i$. A standard Bayesian lower bound for location models implies

$$\sup_{x_i \in \mathbb{R}^d} \mathbb{E}\big[\|x_i - m(Y_i)\|_2^2\big] \ \ge \ \mathbb{E}\|Z_i\|_2^2.$$

Combining the inequalities,

$$\sup_X \mathbb{E}\big[\|x_i - \hat{x}_i(\tilde{X}, W)\|_2^2\big] \ \ge \ \mathbb{E}\|Z_i\|_2^2 = r^2\,T_i.$$

This completes the proof. □

While Theorem 1 provides a tight minimax lower bound on the MSE, it does not prescribe how to choose the noise norm $r$. Our goal is to select $r$ such that encoded samples do not leak meaningful directional information about the originals. A scalar energy ratio alone is insufficient for this purpose, since interference energy could in principle be concentrated in a small subset of coordinates. To address this, we control the *directional* signal-to-noise ratio, defined for every unit vector $u$ as the ratio between the expected squared projection of the signal and that of the interference along $u$. This stronger notion rules out anisotropic interference patterns and ensures that no direction in feature space remains weakly perturbed.

Motivated by the defense rationale in Luo et al. (2022), where distinct augmented images are used to prevent redundancy in mixup inputs, we choose $r$ proportional to the typical separation between data points. Theorem 2 leverages the isotropy of the spherical noise in Algorithm 1 to show that the interference has uniform variance in every direction, and specifies how to select the scaling parameter $m_f$ and thus set $r$ as a multiple of the average inter-sample distance—so that the worst-case directional SNR falls below a prescribed threshold $\tau$.

**Theorem 2.** *Let $\{x_i\}_{i=1}^n \subset \mathbb{R}^d$ be i.i.d. samples of a random vector $X$ with*

$$\mathbb{E}[X] = 0, \qquad V := \mathbb{E}\|X\|_2^2 < \infty.$$

*Let $X'$ be an independent copy of $X$. Algorithm 1 outputs*

$$\tilde{x}_i = w_{i1}x_i + (1 - w_{i1})\big(x_{\pi(i)} + e_i\big), \qquad 0 \le w_{i1} \le \alpha,\ \alpha \in (0,1),$$

*where $w_{i1}$ is independent of the data and*

$$e_i \sim \mathrm{Uniform}(\mathbb{S}(0, r)), \qquad r = m_f D,$$

*with*

$$D := \mathbb{E}\|X - X'\|_2, \qquad c := \frac{D^2}{2V} \in (0,1].$$

*Define*

$$S_i := w_{i1}x_i, \qquad I_i := (1 - w_{i1})\big(x_{\pi(i)} + e_i\big).$$

*For any unit vector $u \in \mathbb{R}^d$, define the directional SNR*

$$\mathrm{SNR}(u) := \frac{\mathbb{E}\,\langle S_i, u\rangle^2}{\mathbb{E}\,\langle I_i, u\rangle^2}.$$

*Then for every* $\|u\|_2 = 1$,

$$\mathrm{SNR}(u) \;\leq\; \frac{\alpha^2\,\mathbb{E}\langle X, u\rangle^2}{(1-\alpha)^2\big(\mathbb{E}\langle X, u\rangle^2 + \frac{r^2}{d}\big)} \;\leq\; \frac{\alpha^2}{(1-\alpha)^2\big(1+\frac{r^2}{dV}\big)}.$$

*Consequently, a sufficient condition for*

$$\sup_{\|u\|_2=1}\ \mathrm{SNR}(u) \leq \tau$$

*is*

$$r^2 \;\geq\; dV\Big(\frac{\alpha^2}{\tau(1-\alpha)^2} - 1\Big),$$

*and using* $r = m_f D$ *and* $D^2 = 2cV$,

$$m_f \;\geq\; \sqrt{\frac{d}{2c}\Big(\frac{\alpha^2}{\tau(1-\alpha)^2} - 1\Big)}.$$

*Proof.* Fix any unit vector $u$.

Since $0 \leq w_{i1} \leq \alpha$ and $w_{i1}$ is independent of $x_i$,

$$\mathbb{E}\langle S_i, u\rangle^2 = \mathbb{E}[w_{i1}^2]\,\mathbb{E}\langle X, u\rangle^2 \leq \alpha^2\,\mathbb{E}\langle X, u\rangle^2.$$

For the interference term, using independence and $\mathbb{E}[X'] = 0$,

$$\mathbb{E}\langle I_i, u\rangle^2 = \mathbb{E}[(1-w_{i1})^2]\,\mathbb{E}\langle X' + e_i, u\rangle^2.$$

The cross term vanishes, hence

$$\mathbb{E}\langle X' + e_i, u\rangle^2 = \mathbb{E}\langle X, u\rangle^2 + \mathbb{E}\langle e_i, u\rangle^2.$$

Since $e_i \sim \mathrm{Uniform}(\mathbb{S}(0, r))$,

$$\mathbb{E}[e_i e_i^\top] = \frac{r^2}{d} I_d, \qquad \Rightarrow \qquad \mathbb{E}\langle e_i, u\rangle^2 = \frac{r^2}{d}.$$

Moreover, $(1 - w_{i1}) \geq (1-\alpha)$ almost surely, so

$$\mathbb{E}[(1-w_{i1})^2] \geq (1-\alpha)^2.$$

Thus

$$\mathbb{E}\langle I_i, u\rangle^2 \geq (1-\alpha)^2\left(\mathbb{E}\langle X, u\rangle^2 + \frac{r^2}{d}\right).$$

Taking the ratio yields the stated bound. The uniform bound follows from $\mathbb{E}\langle X, u\rangle^2 \leq V$. The condition on $m_f$ follows by algebra using $r = m_f D$ and $D^2 = 2cV$. $\qquad\square$

The guarantee in Theorem 2 is distributional: the directional SNR is controlled in expectation with respect to the underlying data distribution through the second-moment quantities $V$ and $\mathbb{E}\langle X, u\rangle^2$. As such, it provides an average-case criterion for selecting $r$ (equivalently $m_f$), ensuring that typical samples are sufficiently masked in every direction. It does not, however, preclude the possibility that atypical or high-energy samples—whose norms are not well reflected by these global second-order statistics—may exhibit a larger per-sample directional SNR. To account for such outliers in practice, we incorporate a simple post-processing step: after applying Algorithm 1, transformed samples can be screened using empirical criteria (e.g., sample-wise SNR estimates or perceptual similarity measures), and discarded if they appear overly revealing. This practical safeguard complements the theoretical average-case guarantee.

Finally, although Theorem 1 characterizes the fundamental difficulty of inversion for adversaries without prior information, practical attackers may possess substantial knowledge about the structure or distribution of the underlying data. To bridge this gap, our experimental evaluation assesses the performance of both linear and nonlinear estimators in realistic scenarios, thereby providing a more comprehensive understanding of the scheme's robustness against adversaries capable of exploiting informative priors.

Our mechanism is related in spirit to differential privacy (DP) in that both aim to limit information leakage from released outputs. However, the objectives and guarantees differ fundamentally. Differential privacy provides worst-case indistinguishability guarantees with respect to neighboring datasets, ensuring that the inclusion or removal of a single individual has a provably bounded effect on the output distribution. In contrast, our singularized mixup mechanism is designed to make inversion ill-conditioned under a reconstruction-based threat model. The guarantees established in Theorems 1 and 2 quantify lower bounds on achievable reconstruction error and control directional signal-to-noise ratios in expectation, rather than providing $(\varepsilon, \delta)$-DP guarantees. Moreover, DP mechanisms calibrate perturbations according to worst-case global sensitivity, whereas our approach selects perturbation magnitude using distribution-dependent second-moment statistics to control directional SNR on average. Thus, although both frameworks aim to mitigate information leakage, they rely on distinct formal definitions, adversarial models, and analytical tools Abadi et al. (2016); Geyer et al. (2017).

## 5 Experiments

Our experiments are designed to evaluate both the accuracy loss relative to InstaHide and the attack resilience of our proposed algorithm.

We evaluate our method on MNIST LeCun (1998), CIFAR-10 and CIFAR-100 Krizhevsky et al. (2009), and Tiny-ImageNet Le & Yang (2015), using a unified implementation in PyTorch. To ensure computational efficiency and reproducibility across extensive experimental sweeps, we employ publicly available convolutional backbones pretrained on ImageNet-1K (as provided by `torchvision`) as fixed feature extractors. Specifically, we use ResNet-18 for MNIST and CIFAR-10, and ResNet-50 for CIFAR-100 and Tiny-ImageNet. In all cases, the pretrained models are used strictly as frozen encoders: their weights are not updated, and no fine-tuning is performed on either original or mixed data. We remove the final classification layers and use the resulting convolutional representations as input to a lightweight classifier trained from scratch. All images are resized to a common spatial resolution and normalized consistently with the ImageNet pretraining protocol; grayscale images are converted to three channels when necessary.

The perturbation mechanism of Algorithm 1 is defined and calibrated in image space. For each dataset, we estimate the relevant scale statistics after preprocessing and determine the corresponding image-space noise magnitude in accordance with Theorem 2. To enable efficient training in feature space while preserving this image-space calibration, we empirically measure how image-space perturbations propagate through the fixed backbone by computing the average displacement they induce in the extracted representations. This measured displacement is then used to set the mixing radius during feature-space training. Importantly, the pretrained backbone serves solely as a fixed, publicly available representation to reduce computational cost; it is not part of the privacy mechanism, does not depend on the private data distribution, and does not influence the theoretical security guarantees of the construction.

Beyond utility evaluation, we conduct a comprehensive security analysis against linear reconstruction attacks, which constitute the primary attack model considered in prior work on InstaHide Carlini et al. (2021a); Luo et al. (2022); Chen et al. (2020). Linear attacks are evaluated on all benchmark datasets considered in our study (MNIST, CIFAR-10/100, and Tiny-ImageNet), and additionally at larger scale on CIFAR-5M Nakkiran et al. (2020). Our singularized mixup procedure is analyzed under a realistic threat model in which the adversary has access to the mixing coefficients, reflecting the information typically available in practical collaborative or outsourced training environments.

In addition, we assess robustness against nonlinear reconstruction attacks based on learned generative models. In this setting, the attacker is assumed to have access to a public image dataset used to train the reconstruc-

tion network; in our experiments, Tiny-ImageNet serves as the public data source, while CIFAR-10 plays the role of the private dataset on which reconstruction performance is evaluated.

We further investigate the behavior of our method in collaborative training scenarios where the dataset is partitioned across multiple parties with heterogeneous local distributions. In this setting, we compare singularized mixup against a standard federated baseline under controlled label and size heterogeneity. The goal is to assess both robustness to statistical non-iid effects and scalability as the number of participants increases. We consider varying degrees of label skew and client size imbalance, and evaluate performance in terms of test accuracy under identical architectural and optimization configurations.

Overall, the experimental results demonstrate that our method achieves a favorable privacy–utility trade-off across diverse settings. It provides strong resistance to both linear and learned nonlinear reconstruction attacks, with reconstruction quality rapidly degrading as the privacy parameter $\tau$ decreases, while maintaining high downstream accuracy even under conservatively small $\tau$. Unlike prior mixing-based defenses, our approach admits a principled, distribution-aware calibration of noise through Theorem 2 and preserves nearly the entire dataset after post-processing. Moreover, in collaborative regimes with substantial label and size heterogeneity, singularized mixup not only improves robustness relative to federated baselines but also reduces the attack surface by avoiding iterative gradient exchange. Together, these properties establish our method as a theoretically grounded, empirically robust, and practically scalable solution for privacy-preserving representation learning.

## 5.1 Security

In the following, we examine two distinct attack strategies for recovering the underlying images from their mixed representations. The first is a linear inversion attack, which exploits the known linear mixing process to directly reconstruct the sources. The second is a non-linear reconstruction attack based on U-Net architectures, allowing the adversary to learn a more flexible, data-driven inverse mapping.

The noise norm is chosen according to Theorem 2. In particular, we estimate the average distance between two randomly selected images directly from the data and set the noise level $r$ to be this empirical average multiplied by the scaling factor prescribed by the theorem. This choice places the method in the theoretical regime analyzed in Theorem 2 and ensures that the *expected* directional SNR—taken with respect to the underlying data distribution—is controlled through global second-order statistics of the data.

Theorem 2 provides a principled and distribution-aware method for selecting the parameter $m_f$ (and hence the noise radius $r$) in Algorithm 1, by controlling the directional SNR in expectation with respect to the data distribution. As such, the guarantee is inherently average-case: it ensures that, for typical samples, the added perturbation sufficiently attenuates recoverable signal energy across various directions, but it does not constitute a worst-case per-sample bound. To strengthen our empirical privacy assessment beyond this second-moment, SNR-based analysis, we additionally report perceptual similarity metrics—SSIM and LPIPS—alongside SNR. While SNR captures energy-based distortion, SSIM reflects structural similarity (ranging from 0 to 1, with values above $\sim 0.9$ typically appearing nearly indistinguishable and values below $\sim 0.5$ showing clear structural degradation), and LPIPS measures similarity in a deep feature space aligned with human perception (with values near 0 indicating high perceptual similarity and larger values, e.g., $> 0.3$, corresponding to visibly different images). Together, these metrics complement SNR by capturing how similar reconstructions look structurally (SSIM) and how similar they feel perceptually (LPIPS), providing a more comprehensive evaluation of reconstruction quality Bovik (2010).

Since Theorem 2 provides an average-case guarantee, it does not preclude the existence of rare outliers for which the singularized representation may retain unusually high similarity to the original sample. To mitigate this effect and strengthen our empirical privacy evaluation, we introduce a simple post-processing step: after applying Algorithm 1, we compute the LPIPS distance between each singularized image and its original counterpart, and discard those samples whose LPIPS is below 0.7. This threshold removes representations that remain perceptually too close to the source image. We first quantify the fraction of excluded samples across datasets and privacy levels. Table 1 reports the percentage of singularized images removed for $\tau \in \{1.0, 0.1, 0.01, 10^{-3}, 10^{-4}, 10^{-5}, 10^{-6}\}$.

Table 1: Percentage (%) of singularized images produced by Algorithm 1 that are excluded by the LPIPS-based post-processing step (LPIPS < 0.9). The privacy parameter $\tau$ denotes the directional SNR threshold defined in Theorem 2; smaller $\tau$ corresponds to stronger signal attenuation and larger calibrated noise magnitude.

| Dataset | $\tau = 1.0$ | $\tau = 0.1$ | $\tau = 0.01$ | $\tau = 10^{-3}$ | $\tau = 10^{-4}$ | $\tau = 10^{-5}$ | $\tau = 10^{-6}$ |
|---|---|---|---|---|---|---|---|
| MNIST | 100.00 | 0.00 | 0.00 | 0.00 | 0.00 | 0.00 | 0.00 |
| CIFAR-10 | 0.15 | 0.00 | 0.00 | 0.00 | 0.00 | 0.00 | 0.00 |
| CIFAR-100 | 0.29 | 0.00 | 0.00 | 0.00 | 0.00 | 0.00 | 0.00 |
| Tiny-ImageNet | 14.06 | 0.00 | 0.00 | 0.00 | 0.00 | 0.00 | 0.00 |
| CIFAR-5M | 0.10 | 0.00 | 0.00 | 0.00 | 0.00 | 0.00 | 0.00 |

The excluded fraction is consistently negligible across all datasets and privacy levels. Even for moderately strong security parameters ($\tau$), only an extremely small proportion of singularized images fall below the LPIPS threshold. This indicates that, in practice, the average-case guarantee of Theorem 2 is not compromised by a meaningful population of high-similarity outliers. In particular, for natural image datasets such as CIFAR-10, CIFAR-100, Tiny-ImageNet, and CIFAR-5M, the post-processing step removes only a vanishingly small subset of samples. Thus, the proposed mechanism achieves strong empirical privacy while preserving virtually the entire dataset for downstream training.

**Linear inversion attack.** We follow a gradient-descent–based reconstruction procedure inspired by Luo et al. (2022). In this setting, the adversary is assumed to know the linear mixing matrix and uses this information to guide the recovery process. The loss function is built around a linear reconstruction term that measures the mean squared error between the observed mixtures and the linearly recomposed images obtained by applying the known mixing weights to the current estimates. To stabilize this inversion, the attacker incorporates generic priors on natural images through additional regularization terms: a total-variation penalty that promotes spatial smoothness and an $\ell_2$ penalty that discourages unrealistically large pixel values. Optimization is carried out with Adam, and after each update the recovered images are clipped to a fixed range to maintain plausible pixel values.

The reconstruction performance of the linear attack is evaluated using SNR (in dB), SSIM, and LPIPS. From the attacker's perspective, higher SNR and higher SSIM indicate more successful reconstruction, while lower LPIPS corresponds to greater perceptual similarity to the original image and thus a stronger attack. Results across datasets and noise levels $\tau$ are reported in Table 2. Figure 3 shows the best recovered image for CIFAR-10 under the linear attack.

**Nonlinear (U-Net) attack.** We replace the analytic inversion used in the linear setting with a learned reconstruction model based on a U-Net architecture. The adversary is assumed to have access to a collection of public images, which are used to train a network that maps mixed inputs back to clean sources. To simulate a realistic and reasonably strong threat model, we assume that the attacker is aware of the semantic classes present in the private dataset. Accordingly, when Tiny-ImageNet is used as public data, we restrict it to categories that semantically correspond to those in CIFAR-10, which serves as the private dataset in our evaluation. This ensures that the attacker's public data reflects the same high-level object categories as the private distribution, without granting access to the private images themselves.

A separate U-Net is trained for each value of $\tau$, allowing the attacker to adapt to the corresponding noise magnitude. Training is performed using an $\ell_1$ reconstruction loss together with total-variation regularization.

To assess robustness with respect to model initialization, each nonlinear attack experiment is repeated across five independent random seeds. Reported results correspond to the mean and standard deviation over these runs, thereby reducing the effect of favorable or unfavorable initializations and providing a more stable estimate of reconstruction performance.

Reconstruction quality for the nonlinear attack is summarized in Table 3. As before, SNR (dB) and SSIM are higher-is-better metrics, while LPIPS is lower-is-better. For completeness, we also report the best SNR achieved across seeds for each noise level.

Table 2: Linear attack reconstruction quality across noise levels $\tau$. For SNR (dB) and SSIM, higher values indicate better reconstruction quality; for LPIPS, lower values indicate better perceptual similarity. Results are reported as mean $\pm$ standard deviation over the test set.

| Dataset | $\tau$ | SNR (dB) ↑ | SSIM ↑ | LPIPS ↓ |
|---------|--------|-----------|--------|---------|
| MNIST | $10^0$ | $8.15\pm0.28$ | $0.4945\pm0.0129$ | $0.6453\pm0.0099$ |
| | $10^{-1}$ | $0.07\pm0.13$ | $0.0869\pm0.0083$ | $0.8257\pm0.0108$ |
| | $10^{-2}$ | $-2.79\pm0.08$ | $0.0363\pm0.0060$ | $0.8658\pm0.0107$ |
| | $10^{-3}$ | $-3.65\pm0.08$ | $0.0286\pm0.0053$ | $0.8726\pm0.0106$ |
| | $10^{-4}$ | $-3.91\pm0.09$ | $0.0267\pm0.0051$ | $0.8746\pm0.0102$ |
| | $10^{-5}$ | $-3.99\pm0.08$ | $0.0262\pm0.0050$ | $0.8753\pm0.0105$ |
| | $10^{-6}$ | $-4.01\pm0.08$ | $0.0260\pm0.0050$ | $0.8755\pm0.0106$ |
| CIFAR-10 | $10^0$ | $0.71\pm2.95$ | $0.4707\pm0.0281$ | $0.8374\pm0.0421$ |
| | $10^{-1}$ | $-5.23\pm2.56$ | $0.1855\pm0.0139$ | $0.9614\pm0.0387$ |
| | $10^{-2}$ | $-7.10\pm2.36$ | $0.1279\pm0.0126$ | $0.9826\pm0.0445$ |
| | $10^{-3}$ | $-7.63\pm2.31$ | $0.1146\pm0.0123$ | $0.9884\pm0.0465$ |
| | $10^{-4}$ | $-7.79\pm2.29$ | $0.1109\pm0.0122$ | $0.9902\pm0.0471$ |
| | $10^{-5}$ | $-7.84\pm2.29$ | $0.1099\pm0.0121$ | $0.9907\pm0.0474$ |
| | $10^{-6}$ | $-7.85\pm2.29$ | $0.1096\pm0.0121$ | $0.9909\pm0.0472$ |
| CIFAR-100 | $10^0$ | $1.02\pm2.99$ | $0.4496\pm0.0342$ | $0.8476\pm0.0463$ |
| | $10^{-1}$ | $-4.78\pm2.50$ | $0.1765\pm0.0173$ | $0.9691\pm0.0444$ |
| | $10^{-2}$ | $-6.56\pm2.27$ | $0.1236\pm0.0150$ | $0.9923\pm0.0532$ |
| | $10^{-3}$ | $-7.08\pm2.22$ | $0.1113\pm0.0144$ | $0.9984\pm0.0560$ |
| | $10^{-4}$ | $-7.23\pm2.20$ | $0.1078\pm0.0142$ | $1.0002\pm0.0569$ |
| | $10^{-5}$ | $-7.28\pm2.19$ | $0.1068\pm0.0141$ | $1.0009\pm0.0569$ |
| | $10^{-6}$ | $-7.29\pm2.19$ | $0.1065\pm0.0141$ | $1.0011\pm0.0572$ |
| Tiny-ImageNet | $10^0$ | $1.32\pm2.30$ | $0.4666\pm0.0291$ | $0.7607\pm0.0471$ |
| | $10^{-1}$ | $-4.50\pm1.88$ | $0.1799\pm0.0163$ | $0.9023\pm0.0452$ |
| | $10^{-2}$ | $-6.30\pm1.69$ | $0.1219\pm0.0147$ | $0.9248\pm0.0514$ |
| | $10^{-3}$ | $-6.82\pm1.64$ | $0.1083\pm0.0142$ | $0.9308\pm0.0532$ |
| | $10^{-4}$ | $-6.97\pm1.63$ | $0.1046\pm0.0141$ | $0.9326\pm0.0539$ |
| | $10^{-5}$ | $-7.02\pm1.62$ | $0.1035\pm0.0140$ | $0.9332\pm0.0543$ |
| | $10^{-6}$ | $-7.03\pm1.62$ | $0.1032\pm0.0140$ | $0.9330\pm0.0540$ |
| CIFAR-5M | $10^0$ | $0.86\pm2.69$ | $0.4669\pm0.0255$ | $0.8388\pm0.0390$ |
| | $10^{-1}$ | $-5.08\pm2.29$ | $0.1820\pm0.0151$ | $0.9626\pm0.0351$ |
| | $10^{-2}$ | $-6.95\pm2.08$ | $0.1252\pm0.0144$ | $0.9842\pm0.0408$ |
| | $10^{-3}$ | $-7.49\pm2.03$ | $0.1122\pm0.0140$ | $0.9900\pm0.0429$ |
| | $10^{-4}$ | $-7.64\pm2.02$ | $0.1086\pm0.0139$ | $0.9919\pm0.0434$ |
| | $10^{-5}$ | $-7.69\pm2.01$ | $0.1076\pm0.0138$ | $0.9925\pm0.0436$ |
| | $10^{-6}$ | $-7.71\pm2.01$ | $0.1073\pm0.0138$ | $0.9926\pm0.0437$ |

Table 3: Nonlinear (U-Net) attack on CIFAR-10 across noise levels $\tau$. Each experiment is repeated over five independent random seeds; results are reported as mean $\pm$ standard deviation across seeds. Higher SNR and SSIM indicate better reconstruction quality, while lower LPIPS indicates better perceptual similarity. The last column reports the best SNR (in dB) achieved among the five runs.

| $\tau$ | SNR (dB) ↑ | SSIM ↑ | LPIPS ↓ | Best SNR (dB) |
|--------|-----------|--------|---------|---------------|
| 1 | $7.58\pm3.46$ | $0.7898\pm0.0443$ | $0.4895\pm0.0917$ | 19.35 |
| $10^{-1}$ | $6.98\pm3.29$ | $0.7156\pm0.0458$ | $0.5457\pm0.0823$ | 18.29 |
| $10^{-2}$ | $5.40\pm3.03$ | $0.6992\pm0.0562$ | $0.5798\pm0.0796$ | 16.45 |
| $10^{-3}$ | $2.87\pm2.48$ | $0.6635\pm0.0764$ | $0.6818\pm0.0790$ | 12.13 |
| $10^{-4}$ | $0.10\pm0.98$ | $0.5717\pm0.0845$ | $0.9126\pm0.0736$ | 3.76 |
| $10^{-5}$ | $-0.14\pm0.56$ | $0.6352\pm0.1059$ | $0.8484\pm0.0852$ | 2.47 |
| $10^{-6}$ | $-0.15\pm0.57$ | $0.6373\pm0.1063$ | $0.8706\pm0.0829$ | 2.46 |

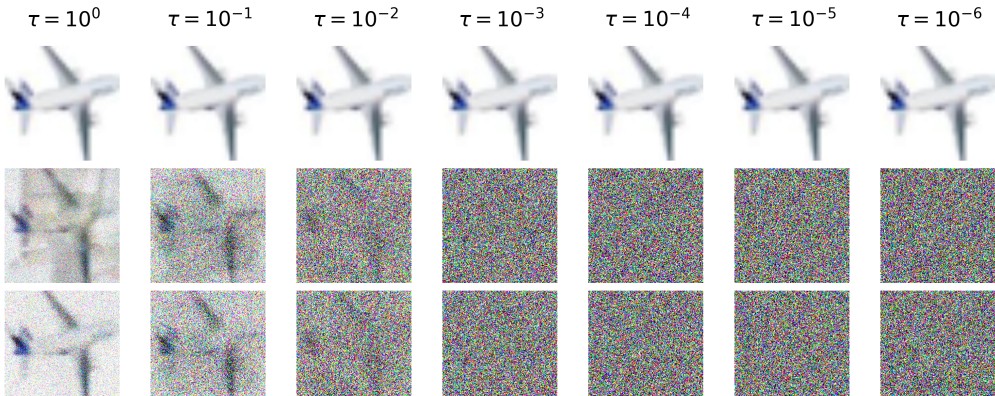

Figure 1: Linear reconstruction attack on CIFAR-10 across decreasing noise levels $\tau$. Columns correspond to different values of $\tau$ (from left to right: $10^0$ to $10^{-6}$). For each column, the top image shows the original private image, the middle image shows the mixed input provided to the attacker, and the bottom image shows the image reconstructed by the linear inversion attack. As $\tau$ decreases (corresponding to stronger noise according to Theorem 2), the recovered images rapidly degrade into noise, illustrating the reduced effectiveness of the attack under stricter privacy settings.

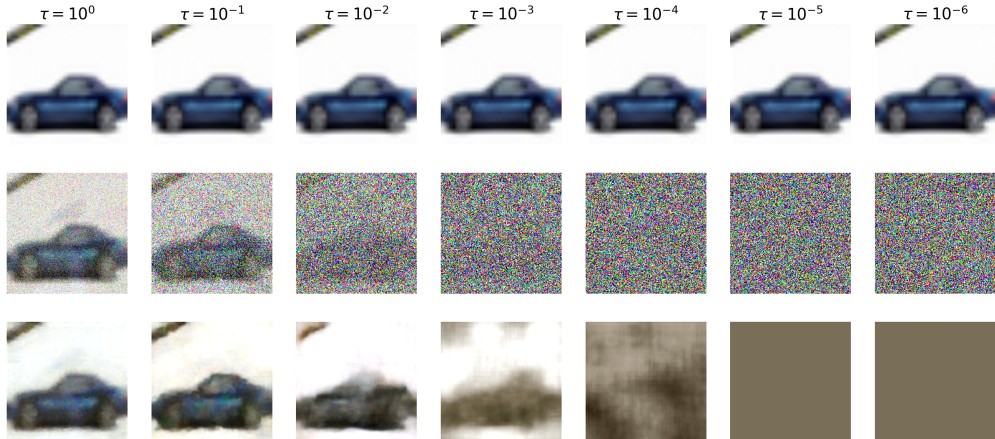

Figure 2: Nonlinear (U-Net) reconstruction attack on CIFAR-10 across decreasing noise levels $\tau$. Columns correspond to different values of $\tau$ (from left to right: $10^0$ to $10^{-6}$). For each column, the top image shows the original private image, the middle image shows the mixed input available to the attacker, and the bottom image shows the reconstruction produced by the trained U-Net. For larger $\tau$ (weaker noise), the nonlinear model is able to recover substantial semantic structure. As $\tau$ decreases (stronger noise), reconstruction quality progressively deteriorates, and the recovered images eventually lose identifiable content, illustrating the increasing robustness of the proposed method against learned inversion attacks.

## 5.2 Accuracy

The security analysis indicates that, although the nonlinear attack is noticeably more powerful than the linear one for $\tau \geq 10^{-4}$, the original image remains unrecoverable across all datasets even at this relatively small noise level. Since $\tau = 10^{-4}$ already suffices to prevent meaningful reconstruction, we adopt a conservative stance and conduct all accuracy experiments at an even stricter privacy setting, using $\tau = 10^{-6}$ uniformly across datasets. At this noise level, we train a standard convolutional classifier on mixed representations and report the resulting test accuracy for each dataset. We then compare these results with the best-performing configuration of InstaHide (with $k = 4$) to quantify the utility–privacy trade-off under a conservative noise regime. The full accuracy results are summarized in Table 4.

Table 4: Test accuracy (%) at $\tau = 10^{-6}$ compared with the best reported InstaHide configuration ($k = 4$).

| Method | MNIST | CIFAR-10 | CIFAR-100 | Tiny-ImageNet |
|---|---|---|---|---|
| InstaHide ($k = 4$) | 99.66 | 91.20 | 74.01 | – |
| Ours ($\tau = 10^{-6}$) | 99.32 | 90.51 | 75.99 | 72.50 |

The accuracy results in Table 4 show that, even under the conservative noise setting $\tau = 10^{-6}$—significantly stricter than what is needed to prevent both linear and nonlinear reconstruction—the loss in predictive performance remains negligible across all datasets. On MNIST and CIFAR-10, our approach matches the accuracy of InstaHide with $k = 4$, and on CIFAR-100 and Tiny-ImageNet it even yields modest improvements. These findings indicate that strong reconstruction resistance does not come at the expense of meaningful degradation in downstream utility: despite enforcing a noise level far below the threshold where the nonlinear attacker fails (i.e., $\tau \approx 10^{-4}$), the mixed representations still support high classification accuracy. Overall, the results demonstrate that our method provides robust security guaranties while preserving competitive model performance across diverse datasets.

## 5.3 Collaborative training

All collaborative experiments are conducted on CIFAR-10 using the same preprocessing and feature-extraction pipeline described earlier. Images are resized and normalized according to the ImageNet pre-training protocol, and a fixed, pretrained ResNet-18 backbone is used as a frozen feature extractor. The perturbation mechanism of Algorithm 1 is calibrated in image space after preprocessing, and the resulting image-space noise magnitude—determined via Theorem 2—is mapped to an equivalent mixing radius in feature space by measuring the induced displacement in the backbone representations. The backbone remains fixed throughout and serves solely as a computationally efficient public representation. In all collaborative experiments, we set the privacy parameter to $\tau = 10^{-6}$, corresponding to the conservative privacy regime used in our main accuracy evaluation.

We next study performance in a collaborative training regime, where the CIFAR-10 training set is partitioned across $P$ parties, each holding a disjoint local subset of equal size. To systematically control statistical heterogeneity, we adopt Dirichlet-based label partitioning: for each class $c$, class proportions across parties are sampled from a symmetric Dirichlet distribution $\mathrm{Dir}(\beta\mathbf{1})$, and examples are allocated accordingly. Smaller values of $\beta$ induce stronger label skew and more pronounced non-iid local datasets. We report results for $P \in \{10, 50, 100\}$ and $\beta \in \{1.0, 0.5, 0.1\}$.

We compare two collaborative training strategies under the same architectural and optimization setup. As a federated baseline, we implement FedProx Li et al. (2020) with 50 communication rounds (one round per epoch). In each round, all parties perform one local training epoch starting from the current global classifier, and the server aggregates the updated local models using data-size-weighted averaging. The FedProx proximal coefficient is set to $\mu = 10^{-3}$ in all experiments.

As our method, we use singularized mixup. Each party independently generates a mixed feature dataset using Algorithm 1, where mixing is performed strictly within-party and the noise magnitude is calibrated locally from the privacy parameter $\tau$. The server then trains a single classifier on the union of all mixed representations (with corresponding soft labels). In contrast to FedProx, singularized mixup is a one-shot collaboration protocol: it does not require iterative communication rounds and does not involve the exchange of gradients or intermediate model updates.

Table 5 reports test accuracy. Across all settings, singularized mixup outperforms FedProx, with the improvement generally increasing as the number of parties grows or as the data become more heterogeneous (smaller $\beta$). For example, under strong label skew ($\beta = 0.1$), singularized mixup improves accuracy by +6.04% for $P = 10$ and by +6.52% for $P = 50$. For larger collaborations, the gap becomes even larger (e.g., +11.83% at $P = 100$, $\beta = 1.0$). These results indicate that aggregating mixed representations provides a robust training signal in regimes where federated optimization can be sensitive to small local sample sizes and client drift.

In addition to accuracy, the protocols differ in their attack surface. FedProx (and more broadly FL) repeatedly transmits gradients or model updates, which can enable gradient-based reconstruction attacks such as gradient inversion Geiping et al. (2020). In contrast, our protocol avoids communicating per-round updates entirely and shares only singularized mixed representations, thereby reducing exposure to known federated gradient leakage vectors.

Table 5: Collaborative test accuracy (%) under Dirichlet label partitioning. FedProx uses iterative communication rounds with local training and server aggregation; singularized mixup trains on the union of party-local mixed representations produced by Algorithm 1. $\Delta$ is the absolute accuracy improvement of singularized mixup over FedProx.

| # Parties $P$ | Dirichlet $\beta$ | FedProx | singularized mixup (Ours) | $\Delta$ |
|---|---|---|---|---|
| 10 | 1.0 | 87.50 | 88.38 | +0.88 |
| 10 | 0.5 | 87.60 | 88.01 | +0.41 |
| 10 | 0.1 | 75.10 | 81.14 | +6.04 |
| 50 | 1.0 | 82.93 | 87.41 | +4.48 |
| 50 | 0.5 | 83.39 | 84.26 | +0.87 |
| 50 | 0.1 | 75.21 | 81.73 | +6.52 |
| 100 | 1.0 | 76.42 | 88.25 | +11.83 |
| 100 | 0.5 | 76.60 | 88.33 | +11.73 |
| 100 | 0.1 | 77.35 | 82.18 | +4.83 |

We further evaluate robustness under *size heterogeneity*, motivated by scenarios where one participant has substantially less data than the rest. Concretely, we consider collaborations with $P = 50$ parties in which a single party (fixed index) is *data-poor* while the remaining $P - 1$ parties are *data-rich* and have equal local dataset sizes. We parameterize this imbalance by a ratio $\rho \in (0, 1]$, defined so that the data-poor party has $n_{\text{poor}} = \rho\, n_{\text{rich}}$ training examples, where $n_{\text{rich}}$ is the per-party dataset size of each data-rich party. We keep the total number of training examples fixed, i.e.,

$$n = (P-1)n_{\text{rich}} + n_{\text{poor}} = (P - 1 + \rho)\, n_{\text{rich}}, \qquad (5)$$

so that varying $\rho$ isolates the effect of client size imbalance rather than changing the overall data budget.

To jointly model label skew and size skew, we extend Dirichlet label partitioning as follows. For each class $c$, we first draw class proportions across parties from $\text{Dir}(\beta \mathbf{1})$ as above, and then bias the resulting allocation toward the desired party sizes using capacity weights: the data-poor party is assigned weight $\rho$ and each data-rich party weight 1. Class-$c$ examples are then assigned to parties according to these reweighted proportions, followed by a balancing step that enforces the target per-party dataset sizes exactly. This procedure preserves the role of $\beta$ as a label-skew control while introducing $\rho$ as an independent knob for size heterogeneity. To avoid degenerate local datasets, we additionally enforce a small minimum number of samples per party.

Table 6 reports collaborative test accuracy for $P = 50$ parties under joint label skew ($\beta \in \{1.0, 0.5, 0.1\}$) and size skew ($\rho \in \{1.0, 0.1, 0.01\}$), where one party is data-poor and the total training set size is fixed.

Across all settings, singularized mixup consistently outperforms FedProx. The improvement ranges from +4.15 to +7.81 percentage points, with the largest gains typically observed under moderate or strong size imbalance ($\rho = 0.1$). While FedProx tends to degrade as client size heterogeneity increases, singularized mixup remains comparatively stable across $\rho$. These results indicate that aggregating mixed feature representations provides robustness to joint label and size heterogeneity.

Table 6: Collaborative test accuracy (%) with one data-poor party under joint label skew ($\beta$) and size skew ($\rho$). We fix $P = 50$ and set $n_{\text{poor}} = \rho \, n_{\text{rich}}$, keeping the total training set size $N$ fixed. $\Delta$ is the absolute accuracy improvement of singularized mixup over FedProx.

| # Parties $P$ | Dirichlet $\beta$ | Size ratio $\rho$ | FedProx | singularized mixup (Ours) | $\Delta$ |
|---|---|---|---|---|---|
| 50 | 1.0 | 1.0 | 83.09 | 88.27 | 5.18 |
| 50 | 1.0 | 0.1 | 81.56 | 88.67 | 7.11 |
| 50 | 1.0 | 0.01 | 81.27 | 87.22 | 5.95 |
| 50 | 0.5 | 1.0 | 83.54 | 88.04 | 4.50 |
| 50 | 0.5 | 0.1 | 81.51 | 89.05 | 7.54 |
| 50 | 0.5 | 0.01 | 81.70 | 88.56 | 6.86 |
| 50 | 0.1 | 1.0 | 82.07 | 86.22 | 4.15 |
| 50 | 0.1 | 0.1 | 80.55 | 88.36 | 7.81 |
| 50 | 0.1 | 0.01 | 80.23 | 85.08 | 4.85 |

## 6 Conclusions

In this paper, we introduced a singularized mixup mechanism that mixes only two private images at a time while injecting structured noise into the non-target component. This design directly addresses the key vulnerability exploited in attacks on InstaHide, where repeated use of the same private image across many mixtures enables clustering and subsequent reconstruction. By corrupting all but one component in each mixture, our method limits what an adversary can infer from any single mixed sample.

We analyzed security under a conservative threat model in which the attacker has full knowledge of the mixing weights and evaluated both a linear inversion attack and a more powerful nonlinear U-Net–based attack. Our experiments show that once the noise magnitude exceeds a modest threshold, neither attack can recover the original image, even when given maximal information about the mixing process. At the same time, using a conservative setting of $\tau = 10^{-6}$—well below the level at which both attacks already fail—the mixed representations retain high utility, with only negligible accuracy loss compared to InstaHide's best $k = 4$ configuration.

While the method is tailored to image data, the singularization principle may extend to other modalities with appropriate noise models. Exploring such extensions presents an interesting direction for future work toward broadly applicable, attack-resistant data mixing schemes. Finally, although our empirical results indicate strong privacy protection under the evaluated threat models, any real-world deployment involving sensitive data (e.g., medical imaging) should undergo domain-specific validation and legal review. As with other data protection mechanisms, practical use must comply with applicable regulatory frameworks such as the General Data Protection Regulation (GDPR) Regulation (2016) and the Health Insurance Portability and Accountability Act (HIPAA) Act et al. (1996), and should be validated by the responsible legal and compliance entities prior to deployment.

## Reproducibility Statement

All source code used in this study is publicly available, along with detailed instructions to reproduce the experiments described in this paper. The data utilized comes from publicly accessible benchmark datasets. For additional details regarding the experimental setup and procedures, please refer to the Appendix E.

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

## A Chen attack

The attack described in Chen et al. (2020) simplifies the InstaHide problem by assuming that the matrix $X \in \mathbb{R}^{d \times n}$ of images is Gaussian, i.e., its entries are chosen i.i.d. from $\mathcal{N}(0,1)$. Let $p_1, \ldots, p_d \in \mathbb{R}^n$ be the rows of $X$. Consider $w_{i1}, dots, w_m \in \mathbb{R}^n$ the unknown selection vectors chosen from a distribution $\mathcal{D}$. $S \subset \{1, \ldots, m\}$ be the coordinates of the public images and $S^c = \{1, \ldots, n\}$ $S$ be the coordinates of the

private images. Let $[v]_S \in \mathbb{R}^{|S|}$ be the restriction of a vector $v$ to the coordinates indexed by $S$. Each selection sector generates an encoded image as:

$$\tilde{x}_i = |Xw_i| \tag{6}$$

In InstaHide, the sign of each pixel from an encoded image is randomly flipped, but as the authors remark, the two notations are interchangeable.

The attack goes like follows:

1. **Learning the public coordinates of any selection vector** In the first step, the attacker determines the weights associated with the public images from each selection vector. Considering the matrix

   $$N = \mathbb{E}_{p,\tilde{x}} \left[ \tilde{x}^2 \cdot ([p]_S [p]_S^{\mathsf{T}} - \mathrm{Id}) \right]$$

   where $\tilde{x} = |\langle w, p \rangle|$, $p \sim \mathcal{N}(0, \mathrm{Id})$. It can be proven that $N$ is a rank-1 matrix proportional to $[w]_S [w]_S^{\mathsf{T}}$. Moreover, $N$ can be approximated by

   $$\hat{N} = \frac{1}{d} \sum_{i=1}^{d} \tilde{x}_i^2 \cdot ([p_i]_S [p_i]_S^{\mathsf{T}} - \mathrm{Id}).$$

2. **Recovering the Gram Matrix** Since the previous steps recover the coordinates of the public images in each selection vector, for the simplicity of the description, consider that all images are private, i.e., $S^c = \{1, \ldots, n\}$. Consider the matrix $\tilde{X} \in \mathbb{R}^{m \times d}$ where each line is an encoded images:

   $$\tilde{X} = \begin{pmatrix} |\langle p_1, w_1 \rangle| & \cdots & |\langle p_d, w_1 \rangle| \\ \vdots & \ddots & \vdots \\ |\langle p_1, w_m \rangle| & \cdots & |\langle p_d, w_m \rangle| \end{pmatrix} \tag{7}$$

   We can use the columns of $\tilde{X}$ to estimate the covariance matrix $\tilde{M}$ of the folded Gaussian distribution $\mathcal{N}^{\mathrm{fold}}(0, M)$, since each column is drawn independently from this distribution. The covariance matrix $M$ is, in fact, the rescaled $m \times m$ Gram matrix whose entries are proportional to the dot product of any two selection vectors; that is, the element at position $(i, j)$ in the matrix $M$ is given by $k \cdot \langle w_i, w_{ij} \rangle$.

3. **Floral submatrices**

   The previous step of the attack shows the dot product between any two selection vectors, i.e., $\langle w_i, w_{ij} \rangle$, thus the attacker knows how many private images are common between two encoded images. In order to identify which private images are common (not only how many), the attacker identifies in $M$ floral submatrices. The rows/columns of a floral submatrix can be indexed by all subsets of size $k$ of a set of $k + 2$ elements where its entries are the intersection sizes between the subsets. More intuitively, the attacker exploits the fact that the subsets of size $k$ of of the set $\{1, \ldots, k + 2\}$ are uniquely identified by their pairwise intersection sizes.

4. **Determining the private images**

   Suppose the attacker has identified a floral matrix in the previous steps, which corresponds to the selection vectors $w_{i1}, \ldots, w_{i_t}$, where $t = \binom{k+2}{k}$. The structure of the floral matrix encodes information about the indices of private images that are common between pairs of selection vectors. Specifically, the row and column indices of the matrix indicate which private images are shared. This allows the attacker to construct a system of equations of the form $|\langle w_{ij}, p_l \rangle| = \tilde{x}_l$ for all $l \in 1, \ldots, d$, where $p_l$ denotes the private images and $\tilde{x}_l$ are known quantities.

   From another perspective, each row or column of the floral submatrix can be indexed by a subset of size $k$ from a set of size $k + 2$. Each element in such a subset represents the index of a private

image. For any given element in the floral matrix—which itself is a submatrix of the Gram matrix $M$—the position of the element along the rows provides the attacker with a set of $k$ private image indices, while the position along the columns provides another set of $k$ indices. By intersecting these two sets, the attacker can determine which private images are common between the selection vectors associated with the corresponding row and column. Solving the resulting system of equations enables the attacker to recover the indices of the private images.

## B  Luo attack

In Luo et al. (2022), the authors observed that the method proposed by Carlini et al. (2021a) can be mitigated by applying data augmentation before the mixup process. To address this, they introduce a new approach that successfully bypasses this mitigation strategy. Their method operates as follows:

1. In the first step, the attacker computes the absolute value of each pixel in every encoded image.

2. Next, a similarity score is calculated for every pair of encoded images to determine, with high probability, whether a given pair is derived from the same private image. To compute this score, the authors propose a comparative network that takes as input both high-resolution and low-resolution versions of the image pairs. This approach yields better results than the standard ResNet architecture used by Carlini. Based on the similarity scores, the attacker clusters the encoded images, with each cluster corresponding to a distinct private image.

3. For each cluster obtained in the previous step, the attacker re-weights all encoded images using the weights associated with the corresponding private image. These weights can be easily inferred from the associated encoded labels. Subsequently, a neural network is trained to perform image relaxation and fusion. This strategy counteracts the effects of geometric image augmentation by generating a set of features that are invariant to geometric transformations. An initial version of the private image is then constructed in the fusion step by combining these feature maps.

4. In the final step, the attacker trains an additional neural network to denoise the image produced in the previous stage.

## C  Carlini attack

The attack consists of two main stages. In the first stage, the attacker determines the two private images used to generate each encoded image during the mixing process. In the second stage, the attacker reconstructs the private images by solving a noisy linear system of equations:

1. The attacker computes the absolute value of each mixup encoding to counteract the random sign changes introduced by the mask $\sigma_i$ in (3).

2. To identify whether two encoded images share at least one common private image, the attacker calculates a similarity function between each pair of encoded images. This similarity function is approximated using a neural network trained on public data transformed via the mixup algorithm. Using the similarity scores, the attacker constructs a weighted graph where vertices represent encoded samples, and edge weights correspond to the similarity function's output.

3. Based on the weighted graph, the attacker identifies densely connected cliques, enabling clustering of encoded samples that share a common private image. Each cluster is represented as a set $S_i$, $1 \leq i \leq n$, where each set contains encoded samples derived from the same private image.

4. Since each encoded image is generated by mixing two private images, the attacker constructs a bipartite similarity graph connecting encoded images to the sets identified in the previous step. Edge weights represent the distance between an encoded image $x_i$ and a set $S_i$. This step determines, for each encoded image, the two sets corresponding to the private images used in its construction.

5. Using the bipartite graph, the attacker maps each encoded image to two sets, representing the private images involved in its generation during the mixup process.

6. The attacker recovers the weights used to generate each encoded image by analyzing the mixup of the labels, as described in (4). Since the labels are one-hot encoded, recovering the associated weights is straightforward.

7. Finally, the attacker constructs a matrix $B \in \mathbb{R}^{n \times d}$, where each row corresponds to an encoded image $\tilde{x}_i$, i.e., $B_i = \tilde{x}_i$. A sparse matrix $M \in \mathbb{R}^{n \times n}$ is also constructed, where each row contains two non-zero entries representing the weights $w^i 1$ and $w_2^i$ associated with the private images used to compute the corresponding encoded image. Let $A \in \mathbb{R}^{n \times d}$ represent the matrix of private images, where each row $A_i = x_i$. The attacker solves the noisy linear system $B = M \cdot A + e$, where $e$ represents the public images used in the mixup. This system can be efficiently solved using gradient descent.

## D  More related work

Liu et al. (2020) proposed a different approach, where a classifier is trained on mixup samples and images to produce mixup results that can later be de-mixed. Unlike previous methods, this approach does not involve training on mixup samples followed by inference on original data. Instead, both training and inference are performed on mixup data, with the inference process generating mixup results that can then be used to recover the correct labels.

In a more recent study, Wang et al. (2024) proposed a mixup-like approach to mitigate model inversion attacks on face recognition systems. Instead of mixing images directly, the authors suggested mixing samples in the frequency domain. Additionally, they employed a reinforcement learning strategy to dynamically determine the number of images to mix, balancing privacy and utility. Similarly, Xiang et al. (2023) introduced a mixing strategy to preserve image privacy during training. Their method involves splitting each image into multiple blocks and replacing parts of these blocks with corresponding blocks from other images with the same label. In another study, Li et al. proposed a new privacy metric called Visual Feature Entropy (VFE), calculated for a region of an image as the sum of squared gradients with respect to both axes. This metric aims to quantify the amount of information that needs protection by analyzing the entropy of a region. The authors' mixing strategy involves shuffling pixels within an image based on the VFE metric. Although this method does not involve computing a weighted sum, it can be interpreted as a form of intra-image data mixing. Eloul et al. (2024) present the concept of mixing gradients in federated learning to enhance security against gradient inversion attacks. Although their method does not involve using random weights for gradient mixing, their straightforward approach of directly averaging gradients across a batch, combined with modifications to the loss function, significantly improves resistance to gradient inversion attacks.

The concept of data mixing is rooted in the broader idea of learnable obfuscation, which encompasses techniques designed to transform data in a way that allows algorithms to learn from the transformed data while safeguarding the privacy of the original data He et al. (2020); Yala et al.; Taki & Mastorakis (2024); Popescu et al. (2022); Nythia et al. (2017). For instance, in Nythia et al. (2017), the authors propose using the Arnold transformation to scramble images before inputting them into a face recognition system. This transformation rearranges image pixels by mapping each pixel to a new location determined by a linear transformation.

In Popescu et al. (2022), a method combining Variational Autoencoders (VAEs) with a substitution technique is introduced to protect medical images during neural network analysis. The approach involves training a VAE to reconstruct the image and then applying a substitution table to the latent space representation of the data. Similarly, Taki & Mastorakis (2024) presents a method to ensure the privacy of both training data and neural network architecture. For image data, the authors propose transforming it into a higher-dimensional space. To protect the architecture, they introduce random subnetworks with synthetic parameters that do not affect the network's accuracy or data flow.

The NeuraCrypt method, proposed in Yala et al., protects data privacy by transforming it with a random neural network. This approach is extended to enable privacy-preserving collaborative training, where all

parties share transformed data with a central server. For the server to learn patterns from the combined datasets, all parties must use the same neural network for data transformation. Finally, He et al. (2020) introduces a privacy-preserving method that applies a linear transformation to each data sample. The authors also provide formal proofs demonstrating the information-theoretic security of their approach under specific conditions.

A common characteristic of learnable obfuscation techniques is that the same transformation—though potentially generated using independently chosen random parameters—must be applied to all samples in the dataset being protected. This creates a notable vulnerability: such techniques cannot provide security against chosen-plaintext attacks. This limitation, formally introduced and proven in Xiao et al. (2024), highlights an inherent weakness in these methods. Informally, learnable obfuscation can protect the privacy of plaintext data only under the assumption that the attacker does not have prior knowledge of the original data.

At first glance, this assumption may seem reasonable, as protecting data already known to an attacker might appear unnecessary. However, in practical scenarios, this assumption often fails. For instance, to improve the generalization capabilities of a machine learning model, private datasets are frequently augmented with publicly available data. For example, a private image dataset might be enriched with images from CIFAR-100 Krizhevsky et al. (2009). To preserve the discriminative properties of the data and enable the model to generalize, the added public data must undergo the same transformation used to protect the private dataset.

This practice introduces a significant risk: an attacker with access to both the original public dataset and its transformed version could potentially design algorithms to reverse-engineer the transformation applied to the private data. An example of this vulnerability is described in Carlini et al. (2021b), where the authors successfully developed an algorithm to solve the NeuraCrypt challenge Yala et al., effectively bypassing the intended privacy protections.

## E  Experimental Details

All experiments were developed using the PyTorch framework and performed on an NVIDIA L4 GPU with 24 GB of available VRAM. Across all datasets, we consistently used a batch size of 128. For optimization, we employed the `AdamW` optimizer with a weight decay of $1 \times 10^{-4}$. The initial learning rate was set to 0.001 for all benchmarks, and we utilized a cosine annealing learning rate scheduler.

Our experimental evaluation was conducted on three distinct benchmarks using ResNet architectures (He et al., 2016), with specific configurations detailed in Table 7. A key aspect of our methodology is the exclusive use of the feature extraction layers from these architectures; the final classifier layers were omitted as our focus is on training features. For all datasets we apply similar transformations, which consist of a resize operation (224 pixels height and width) and a normalization. The MNIST dataset is also adjusted such that it has three channels, making it compatible with the chosen architectures.

Table 7: The configurations used within experiments for each dataset. The mean and standard deviation values for MNIST are for a single channel, while for CIFAR datasets they correspond to the (R, G, B) channels.

| Dataset | Architecture | Epochs | Mean | Std. |
|---|---|---|---|---|
| MNIST | ResNet18 | 120 | 0.1307 | 0.3081 |
| CIFAR-10 | ResNet18 | 200 | [0.4914, 0.4822, 0.4465] | [0.2470, 0.2435, 0.2616] |
| CIFAR-100 | ResNet50 | 200 | [0.5071, 0.4867, 0.4408] | [0.2675, 0.2565, 0.2761] |
| Tiny-ImageNet | ResNet50 | 200 | [0.5071, 0.4867, 0.4408] | [0.2675, 0.2565, 0.2761] |

The training objective was to minimize the following loss function, which is designed for our Mixup implementation:

$$\mathcal{L} = -\frac{1}{N} \sum_{i=1}^{N} \sum_{j=1}^{C} y_{ij} \log(\text{softmax}(\mathbf{o}_i)_j) \tag{8}$$

where $N$ is the batch size, $C$ is the number of classes, $\mathbf{y}_i$ is the (potentially soft) label for sample $i$, and $\mathbf{o}_i$ is the model output for sample $i$. In PyTorch, this is implemented as:

```
loss = -(labels * torch.log_softmax(outputs, dim=1)).sum(dim=1).mean()
```

For the final classification step, we used a custom feed-forward neural network with three dense layers. The first and second of these dense layers are followed by batch normalization, GELU activation, and then a dropout. The precise structure of this neural network is described by the following equation:

$$
\begin{aligned}
\texttt{Classifier}(x; n_{\text{cls}}) = \text{Flatten}(x) \;&\rightarrow\; \text{Linear}_{in;1024} \;\rightarrow\; \text{BN}_{1024} \;\rightarrow\; \text{GELU} \;\rightarrow\; \text{Dropout}(0.5) \\
&\rightarrow\; \text{Linear}_{1024;512} \;\rightarrow\; \text{BN}_{512} \;\rightarrow\; \text{GELU} \;\rightarrow\; \text{Dropout}(0.5) \\
&\rightarrow\; \text{Linear}_{512;n_{\text{cls}}}
\end{aligned}
\tag{9}
$$

The $n_{\text{cls}}$ term represents the number of classes that the classification must be made on (e.g., 10 for MNIST). The $in$ dimension of the flattened tensor $x$ within the first linear layer is 25088 ($512 \times 7 \times 7$ for ResNet18 and ResNet34).

All collaborative training methods are evaluated using a common global test set corresponding to the underlying dataset (e.g., the standard CIFAR-10 test split). After completion of the collaborative procedure—whether FedProx or singularized mixup—the resulting global classifier is evaluated centrally on this held-out test set. Test accuracy is computed over the full test dataset and is identical for all participating parties, since the learned model parameters are shared and no party-specific fine-tuning is performed at evaluation time. Thus, the reported results reflect global generalization performance rather than client-specific metrics, ensuring a consistent and fair comparison across collaborative strategies.

## F  Attack Implementation Details

All attacks use the same preprocessing described in Sec. E, including resizing inputs to $224 \times 224$, ImageNet-style normalization, and converting MNIST to three channels. Experiments are executed on a single NVIDIA L4 GPU (24 GB) with fixed random seed (0).

For the *linear reconstruction attack*, we optimize a recovery tensor using the PyTorch Adam optimizer for 200 steps with learning rate 0.05. We include anisotropic total variation regularization and an $\ell_2$ penalty with default weights $\lambda_{\text{tv}} = 10^{-3}$ and $\lambda_2 = 10^{-4}$. After each step, recovered tensors are clamped back into the normalized image range. Recovery quality is measured using per-image SNR in dB, reported as mean ± std. A single visualization panel is produced: columns correspond to different $\tau$ values and, for each dataset, three aligned rows display the original, the Mixup input, and the recovered image. The displayed index per dataset is chosen as the best-recovered example at the largest $\tau$.

For the *non-linear attack*, we train a U-Net denoiser on Tiny-ImageNet Mixup pairs and evaluate zero-shot on CIFAR-10. The U-Net follows a three-level encoder–decoder design with base width $B = 48$. The encoder consists of successive DoubleConv blocks with channel progression ($3{\rightarrow}B{\rightarrow}2B{\rightarrow}4B{\rightarrow}8B$), separated by $2 \times 2$ max-pooling. The decoder mirrors this structure with transposed convolutions for upsampling, skip connections from encoder features, DoubleConv blocks with channel progression ($8B{\rightarrow}4B{\rightarrow}2B{\rightarrow}B$), and a final $1{\times}1$ convolution mapping to three output channels. All layers use ReLU activations, and outputs are clamped to the normalized range. Training uses Adam with batch size 32, learning rate $10^{-3}$, 30 epochs, and an $\ell_1$ loss augmented with a small TV penalty ($10^{-4}$). Partner selection is deterministic per index so the same samples align across $\tau$ values. Evaluation again reports SNR (mean ± std). A single summary panel is generated: the first row repeats the ground-truth image associated with the best reconstruction at the largest $\tau$, while the second and third rows show the corresponding Mixup and recovered images for each $\tau$ (displayed as columns). Only the $\tau$ labels appear above columns.

All runs save a single figure per attack configuration along with a timestamped log containing the full console output.

## G Full Figure 3

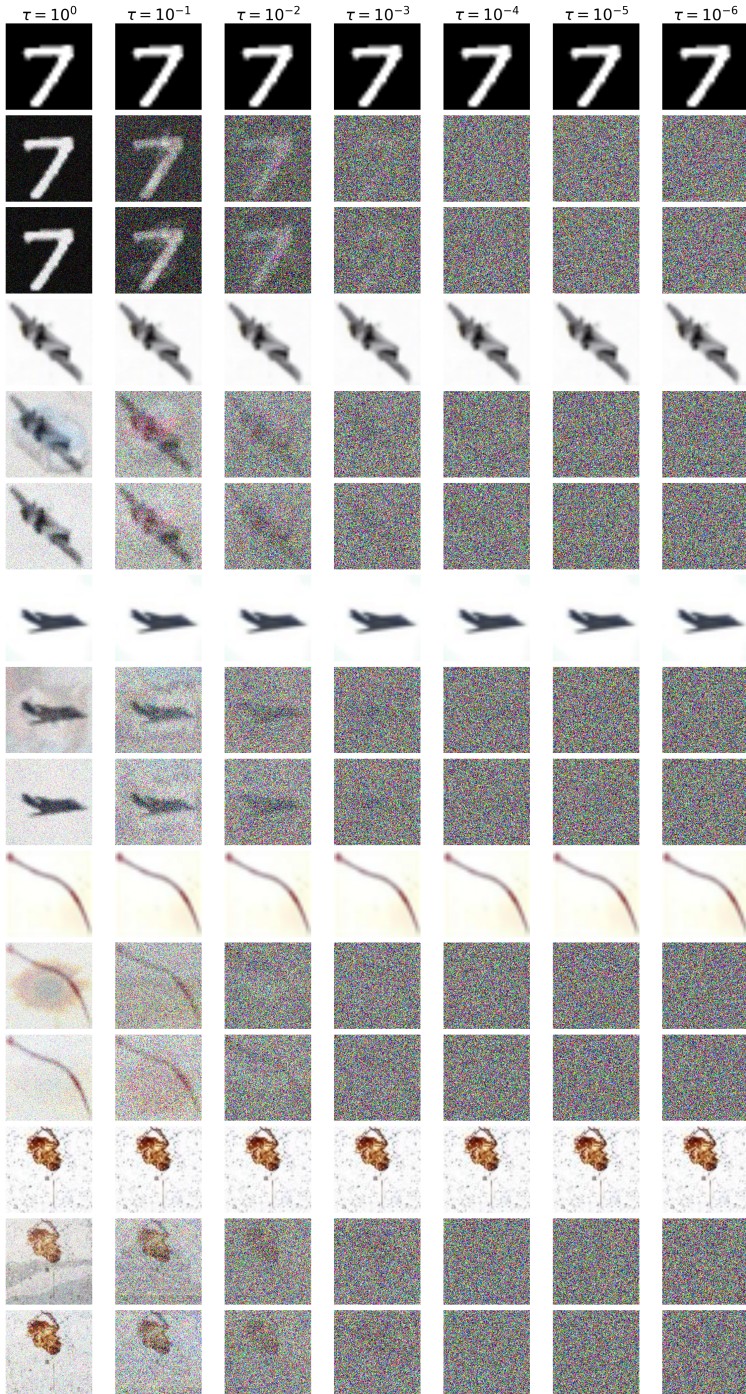

Figure 3: Recovered images under the linear attack for decreasing noise levels $\tau$

## H The use of Large Language Models

We utilized Large Language Models (LLMs) in three specific ways during this work. First, after conducting a manual review of the state of the art using traditional search engines such as Google Scholar, we used

LLMs to assist in identifying additional relevant papers. Second, LLMs were employed to help implement the experiments described in this study. Third, LLMs were used for grammar correction and minor improvements to the flow of the text. Importantly, LLMs were not used to generate or write any paragraphs; their role in writing was limited to minor edits and enhancements.

