# OpenReview forum: "Augmented Mixup Procedure for Privacy-Preserving Collaborative Training"
_TMLR — Accepted by TMLR_

### Review · Reviewer_mEt2 · 2026-02-21

**Summary Of Contributions:**

In this work, the authors build upon the Mixup and Instahide algorithms for privacy-preserving collaborative training. Rather than relying on randomnness of public data or permutations of private data, they propose to add uniformly random vectors, in a data-independent fashion. The authors prove a lower bound on mean squared error of any estimator and upper bound the signal-to-noise ratio. Experiments suggest that the technique can obfuscate images while preserving classification accuracy

**Additional Comments:**

Empirical Privacy experiments: The authors write that "Tiny-ImageNet serves as the public dataset for training the adversary, while CIFAR-10 plays the role of the private dataset used for evaluation." My understanding is that the former has 200 classes while the latter has 10 classes; is it possible for the attacker to make a prediction that is not one of the 10? And would it not be natural to see what happens if the adversary and owner of private data both draw from the same source?

**Audience:**

No

**Audience Explanation:**

I assume TMLR's audience has a passing familiarity with the terms "reconstruction attack", "membership inference attack" and "differential privacy". Under this assumption, I'm not sure what the takeaway of this paper is.
- The form of the pertubation---add noise to (scaled) data---is strikingly similar to that of (local) differential privacy but the initial vector $x_i$ is not bounded so it would seem it's always possible to pick a strong signal for $x_i$ that is greater than fixed $r$ (see "Outliers" above).
- As implied from the counterexample above, the SNR metric is not as compelling as prior ones for data reconstruction or membership inference e.g. reconstruction robustness (Borja Balle et al.)

**Claims And Evidence:**

No

**Claims Explanation:**

I have three concerns:
- Target metric: Low signal-to-noise ratio certainly feels like a necessary condition for a privacy-preserving procedure, but I'm not convinced it's sufficient as a metric of privacy. Consider an alternative to Algorithm 1 where we replace $w_{i2}(x_{\pi i} +e_i)$ with a vector that is all zeroes except for one coordinate where the expected squared value is set to an arbitrary positive number. If that number happens to be $(1-\alpha)^2(V+r^2)$, it looks like the SNR bound in Theorem 2 also applies but the distribution only hides one coordinate of the entire vector.
- Outliers: from my understanding, it seems that the algorithm is good at protecting "average pictures": the proof of Theorem 2 is using the variance of a randomly chosen sample $x_{\pi(i)}$ and the error vector $e_i$ to hide the signal of any given sample $x_i$. But what about atypical pictures? Will the protections still hold if $n-1$ pictures are, say, low intensity grayscale surveillance images but one image is a clear color one? This might happen if a surveillance camera gets upgraded for example
- Empirical Accuracy experiments: The authors write that "feature maps are extracted using a pretrained ResNet-18 model for MNIST and CIFAR-10, and a pretrained ResNet-50 model for CIFAR-100 and Tiny-ImageNet." What is the extent of the pre-training? Is it not possible that the results would change if the images being put through Algorithm 1 are completely different than what is in the pretraining?

**Requested Changes:**

Please discuss prior work mentioned above and address the three main concerns.

Clarify how the entries of Tables 1 and 2 are computed.

Is the label $y_i$ something worth protecting? If it is not, why mix with a random other label? If it is, why not add noise like $e_i$?

Theorem 1 could be simplified: if the authors are already assuming that $x_i$ is the only nonzero contribution, then it seems that it suffices to lower bound $\frac{w_{i2}}{w_{i1}} \cdot e_i$. In that case, we have a lower bound of $(\frac{1-\alpha}{\alpha})^2\cdot r^2$.

---

> ### Author Response · Authors · 2026-03-03
> **Response - 1**
>
> **Regarding Concern 1**
>
> We agree with the reviewer that a low global SNR is a necessary but not sufficient condition for evaluating privacy leakage. In particular, a scalar SNR bound alone does not preclude anisotropic interference patterns in which energy is concentrated in a small subset of coordinates while leaving others weakly perturbed.
>
> In the revised manuscript, we strengthened Theorem 2 accordingly. Instead of controlling only the global energy ratio
>
> $
> \mathrm{SNR} = \frac{\mathbb{E}\|S_i\|_2^2}{\mathbb{E}\|I_i\|_2^2},
> $
>
> we now bound the *directional* SNR
>
> $
> \mathrm{SNR}(u) = \frac{\mathbb{E}\langle S_i,u\rangle^2}{\mathbb{E}\langle I_i,u\rangle^2}
> $
>
> uniformly over all unit vectors $u$. This ensures that the interference variance is uniformly lower bounded in every direction, eliminating constructions where noise energy concentrates in only a few coordinates.
>
> The key ingredient is the isotropy of the noise $e_i \sim \mathrm{Uniform}(\mathbb{S}(0,r))$, which implies
> $
> \mathbb{E}[e_i e_i^\top] = \frac{r^2}{d} I_d.
> $
> Hence, the interference term has variance at least $(1-\alpha)^2 \frac{r^2}{d}$ in every direction. While isotropy was already present in Algorithm 1 through uniform spherical sampling, it was not explicitly used in the original theorem; we now incorporate it directly into the guarantee.
>
> We emphasize that even this strengthened directional SNR bound remains a necessary condition rather than a full characterization of privacy. Theorem 2 is primarily used to select the noise radius $r$ via the scaling parameter $m_f$.
>
> To further substantiate privacy empirically, we extend our evaluation beyond SNR and now report:
>
> - Structural Similarity Index (SSIM),
> - Learned Perceptual Image Patch Similarity (LPIPS).
>
> These complementary metrics assess pixel-level fidelity as well as perceptual and semantic recoverability.
>
> **Regarding Concern 2**
>
> The updated Theorem 2 provides an average-case guarantee: the directional SNR is bounded in expectation through second-moment quantities $V$ and $\mathbb{E}\langle X, u \rangle^2$. We agree with the reviewer that the theorem is inherently an average-case result rather than a worst-case guarantee. Its main purpose is to offer a principled way to calibrate $r$ (equivalently $m_f$) so that typical samples are sufficiently masked.
>
> The reviewer’s outlier example (e.g., $n-1$ low-intensity grayscale images and one clear color image) correctly illustrates that an individual sample may contain significantly more energy than suggested by global second-order statistics. Our current theoretical analysis does not exclude such atypical cases, and we appreciate this insightful observation.
>
> In our collaborative training framework, each party applies Algorithm 1 locally, and training is performed on the union of transformed samples. To mitigate potential outliers in practice, we introduce a per-sample screening step: after singularization, we compute the LPIPS distance between each transformed image $\tilde{x}_i$ and its original $x_i$, and discard samples whose LPIPS is below 0.7. This procedure captures rare cases where a transformed image remains perceptually too similar to the source.
>
> We added a new experiment quantifying the fraction of removed samples across privacy levels $\tau \in \{1.0, 0.1, 0.01, 10^{-3}, 10^{-4}, 10^{-5}, 10^{-6}\}$, where $\tau$ denotes the directional SNR threshold.
>
> Percentage (%) of singularized images excluded by LPIPS < 0.7
>
> | Dataset        | 1.0   | 0.1  | 0.01 | 1e-3 | 1e-4 | 1e-5 | 1e-6 |
> |---------------|-------|------|------|------|------|------|------|
> | MNIST         | 100.00| 0.00 | 0.00 | 0.00 | 0.00 | 0.00 | 0.00 |
> | CIFAR-10      | 0.15  | 0.00 | 0.00 | 0.00 | 0.00 | 0.00 | 0.00 |
> | CIFAR-100     | 0.29  | 0.00 | 0.00 | 0.00 | 0.00 | 0.00 | 0.00 |
> | Tiny-ImageNet | 14.06 | 0.00 | 0.00 | 0.00 | 0.00 | 0.00 | 0.00 |
> | CIFAR-5M      | 0.10  | 0.00 | 0.00 | 0.00 | 0.00 | 0.00 | 0.00 |
>
> Across practically relevant privacy levels, the excluded fraction is negligible. This suggests that while Theorem 2 provides an average-case guarantee, it is not undermined in practice by a significant population of high-similarity outliers, and nearly the entire dataset remains available for training.
>
> The manuscript now explicitly (i) clarifies the average-case nature of Theorem 2, (ii) discusses its limitations for atypical samples, and (iii) incorporates this empirical safeguard. In practice, additional screening mechanisms—such as simple visual inspection or other perceptual similarity checks—can also be employed to further guard against rare pathological cases.

---

> ### Author Response · Authors · 2026-03-03
> **Response - 2**
>
> **Regarding Concern 3**
>
> In our experiments, “pretrained” refers to standard ImageNet-1K pretrained ResNet models (ResNet-18 for MNIST/CIFAR-10 and ResNet-50 for CIFAR-100/Tiny-ImageNet), used strictly as frozen feature extractors. Their weights are never updated and no fine-tuning is performed.
>
> The perturbation mechanism operates entirely in image space. Noise is injected according to Algorithm 1, and its magnitude is calibrated in image space via Theorem 2 (directional SNR bound). Only afterward do we measure how perturbations propagate through the fixed backbone to determine an equivalent mixing radius when training lightweight classifiers on extracted representations.
>
> The pretrained backbone is not part of the privacy mechanism. It does not depend on private data and plays no role in the theoretical guarantees. Reconstruction resistance in Theorems 1–2 arises solely from singularized mixing and isotropic noise injection.
>
> Regarding Tables 2 and 3 (revised numbering): no pretrained backbone is used there. These tables evaluate attack performance directly in image space, consistent with the theoretical setup.
>
> - Table 2: linear attack over image pixels.
> - Table 3: nonlinear attack in image space.
>
> No feature extractor or fine-tuning is involved in these attack evaluations. We clarified this distinction in the revised manuscript.
>
> **Regarding Theorem 1 Simplification**
>
> In the special case where $W$ is diagonal, the bound reduces to a scaled version of the local noise term, yielding $\left(\frac{1-\alpha}{\alpha}\right)^2 r^2$. However, in our setting $W = D_1 + D_2 P_\pi$ includes a permutation component and is generally non-diagonal (unless $\pi$ is the identity). Consequently, $W^{-1}$ is typically dense, and the effective noise affecting coordinate $i$ involves linear combinations of multiple noise terms, requiring the more general bound in Theorem 1.
>
> **Regarding Labels**
>
> In our threat model, labels are not assumed to be private. This is an inherent characteristic of Mixup-based obfuscation techniques, where labels are typically released (either explicitly or implicitly) as part of the training signal. Our goal is therefore not to protect labels, but to design a transformation mechanism whose security does not rely on keeping label-related information secret.
>
> Prior work, such as InstaHide, implicitly relied on the secrecy of mixing weights. However, in Mixup-style methods, mixing weights can often be inferred from the mixed labels themselves. Indeed, previous attacks on InstaHide exploited access to the (public) mixed labels to deduce the mixing weights and subsequently recover the original images. This demonstrates that assuming weight secrecy is unrealistic in practice.
>
> Our objective was to propose a mechanism—Singularized Mixup (Algorithm 1)—and analyze its security under a realistic and conservative threat model. Specifically, we assume from the outset that the attacker has full access to the mixing weights (and labels). Privacy in our framework does not stem from hiding these quantities, but rather from the injected noise and the resulting ill-conditioned inverse problem. By adopting this stronger adversarial assumption, our analysis avoids reliance on obscurity and instead provides guarantees that remain meaningful even when weight and label information is fully exposed.
>
> **Regarding the Public Source of Nonlinear Attack**
>
> Tiny-ImageNet has 200 classes while CIFAR-10 has 10. In the revised experiments, we use Tiny-ImageNet only for classes overlapping with CIFAR-10 (9 common semantic categories). Thus, the adversary is trained on semantically aligned classes, avoiding mismatch and ensuring a consistent evaluation.

---

> ### Author Response · Authors · 2026-03-03
> **Response - 3**
>
> **Regarding TMLR’s audience being interested in the findings of this paper**
>
> We agree that it is important to clarify the practical relevance of our work for the TMLR audience. The main application we target is collaborative training, where multiple parties jointly train a model without sharing their raw data. This setting is central to modern machine learning practice in cross-silo, cross-institutional, or privacy-constrained scenarios.
>
> In the revised paper, we significantly expanded the experimental section to better demonstrate the behavior of singularized mixup in realistic collaborative regimes. In particular, we added extensive experiments on CIFAR-10 under controlled statistical heterogeneity, comparing our approach against the federated baseline FedProx.
>
> First, we study label heterogeneity using Dirichlet partitioning with $P = 10, 50, 100$ parties and $\beta = 1.0, 0.5, 0.1$, where smaller $\beta$ induces stronger label skew. FedProx performs iterative communication with local updates and server aggregation, whereas singularized mixup is a one-shot protocol: each party generates locally mixed representations and the server trains a classifier on the union of these representations, without exchanging gradients or intermediate updates.
>
> The results are summarized below:
>
> | # Parties $P$ | Dirichlet $\beta$ | FedProx | singularized mixup (Ours) | $\Delta$ |
> |---------------|-------------------|---------|---------------------------|----------|
> | 10  | 1.0 | 87.50 | 88.38 | +0.88 |
> | 10  | 0.5 | 87.60 | 88.01 | +0.41 |
> | 10  | 0.1 | 75.10 | 81.14 | +6.04 |
> | 50  | 1.0 | 82.93 | 87.41 | +4.48 |
> | 50  | 0.5 | 83.39 | 84.26 | +0.87 |
> | 50  | 0.1 | 75.21 | 81.73 | +6.52 |
> | 100 | 1.0 | 76.42 | 88.25 | +11.83 |
> | 100 | 0.5 | 76.60 | 88.33 | +11.73 |
> | 100 | 0.1 | 77.35 | 82.18 | +4.83 |
>
> Across all regimes, singularized mixup consistently outperforms FedProx. The performance gap generally increases with the number of parties and under stronger heterogeneity. For instance, under strong label skew ($\beta = 0.1$), the improvement reaches +6.04% for $P = 10$ and +6.52% for $P = 50$, while for large collaborations ($P = 100$) the gain exceeds +11% for $\beta = 1.0$ and $\beta = 0.5$. These results indicate that aggregating mixed representations provides a stable training signal in regimes where federated optimization can suffer from client drift and small local sample sizes.
>
> We further evaluate robustness under joint label and size heterogeneity by considering $P = 50$ parties where one participant is data-poor. We vary the size ratio $\rho = 1.0, 0.1, 0.01$ while keeping the total dataset size fixed, and combine this with Dirichlet label skew ($\beta = 1.0, 0.5, 0.1$). The results are:
>
> | # Parties $P$ | Dirichlet $\beta$ | Size ratio $\rho$ | FedProx | singularized mixup (Ours) | $\Delta$ |
> |---------------|-------------------|-------------------|---------|---------------------------|----------|
> | 50 | 1.0 | 1.0  | 83.09 | 88.27 | 5.18 |
> | 50 | 1.0 | 0.1  | 81.56 | 88.67 | 7.11 |
> | 50 | 1.0 | 0.01 | 81.27 | 87.22 | 5.95 |
> | 50 | 0.5 | 1.0  | 83.54 | 88.04 | 4.50 |
> | 50 | 0.5 | 0.1  | 81.51 | 89.05 | 7.54 |
> | 50 | 0.5 | 0.01 | 81.70 | 88.56 | 6.86 |
> | 50 | 0.1 | 1.0  | 82.07 | 86.22 | 4.15 |
> | 50 | 0.1 | 0.1  | 80.55 | 88.36 | 7.81 |
> | 50 | 0.1 | 0.01 | 80.23 | 85.08 | 4.85 |
>
> Again, singularized mixup consistently outperforms FedProx across all combinations of label and size skew, with gains ranging from +4.15 to +7.81 percentage points. Notably, while FedProx degrades as client size imbalance increases, singularized mixup remains comparatively stable.
>
> We believe these expanded collaborative experiments clarify the practical impact of our approach. The results demonstrate improved robustness to non-iid data and client size heterogeneity, while also avoiding iterative gradient exchange. This combination of theoretical guarantees and strong empirical behavior in realistic collaborative settings makes the findings directly relevant to TMLR’s readership.
>
> We sincerely thank the reviewer for the thoughtful and constructive comments throughout the manuscript. These suggestions have helped us improve the clarity, rigor, and overall presentation of our work.

---

> > ### Comment · Reviewer_mEt2 · 2026-03-14
> > **Thank you + more questions**
> >
> > I thank the authors for their thorough responses to our questions! I think I'm viewing the submission more favorably in light of these changes.
> >
> > That being said, I would like clarification regarding the new work in Section 5.3
> > - Why is $\mu=10^{-3}$? Is this from the prior work? Would results change substantially for other values?
> > - What is the $\tau$ value for singularized mixup? Again, would results change substantially for other values?
> > I ask these because I am very surprised at the outcome of the experiment. I would have thought all the noise being added would surely be more detrimental to accuracy than the configuration of FedProx. Any extra intuition would help here (not just me but readers in general)
> >
> > I also encourage the authors to make explicit reference to differential privacy and compare the standard approaches there (adding noise to mask *bounded* contributions) with the approach taken in Algorithm 1. Even though the objective / threat model in this submission is not the same as in differential privacy, I find the overlap in techniques sufficient to warrant some amount of discussion.

---

> > > ### Author Response · Authors · 2026-03-15
> > > **Response**
> > >
> > > We thank the reviewer for the constructive feedback.
> > >
> > > For FedProx, we chose $\mu = 10^{-3}$, which is part of the standard grid 0.001, 0.01, 0.1 and 1considered in the original FedProx paper (Li et al., 2020). Importantly, Figure 11 in the FedProx paper shows that the main difference in loss appears between $\mu = 0$ and $\mu > 0$: introducing a positive proximal term stabilizes training under heterogeneity. Once $\mu$ is positive, however, using a fixed positive $\mu$ or selecting $\mu$ adaptively does not lead to a large difference in final loss; the main effect is on stability rather than on the attained objective value. This indicates that the dominant qualitative transition is enabling proximal regularization at all, rather than fine-grained tuning among reasonable positive values.
> > >
> > > We therefore view $\mu = 10^{-3}$ as a standard and reasonable fixed choice within the canonical FedProx range. We intentionally keep $\mu$ fixed across all $(\beta, P)$ configurations so that the only varying factor in Section 5.3 is the degree of heterogeneity induced by the partition. While additional tuning of $\mu$ could slightly affect absolute FedProx accuracy, the evidence from the original FedProx study suggests that moderate variation among positive $\mu$ values is unlikely to qualitatively change the comparison between methods.
> > >
> > > Regarding $\tau$, we set $\tau = 10^{-6}$, which corresponds to a stringent security level. Increasing $\tau$ would reduce the perturbation magnitude and move performance closer to the non-perturbed regime. Thus, the reported results should be interpreted as conservative from the utility standpoint. We clarify this explicitly in the revised manuscript.
> > >
> > > Finally, in response to the reviewer’s suggestion regarding differential privacy (DP), we have inserted a paragraph at the end of Section 4 clarifying the distinction. In brief, while both DP and our method aim to mitigate information leakage, DP provides worst-case $(\varepsilon, \delta)$ guarantees via sensitivity-based noise calibration, whereas our approach targets inversion ill-conditioning under a reconstruction-based threat model and controls directional signal-to-noise ratios in expectation.
> > >
> > > References
> > >
> > >
> > > Li, Tian, et al. "Federated optimization in heterogeneous networks." Proceedings of Machine learning and systems 2 (2020): 429-450.

---

> > > ### Author Response · Authors · 2026-03-15
> > > **Regarding the intuition**
> > >
> > > The key distinction is that in Singularized Mixup the perturbation degrades reconstruction but does not disrupt the supervised signal in a coherent way.
> > >
> > > First, cross-entropy is linear in the label. Since the mixed label is a convex combination $\tilde y = w_{i1} y_i + w_{i2} y_{\pi(i)}$, the resulting gradient satisfies $\nabla_\theta \mathcal{L}(f(\tilde x), \tilde y) = w_{i1} \nabla_\theta \mathcal{L}(f(\tilde x), y_i) + w_{i2} \nabla_\theta \mathcal{L}(f(\tilde x), y_{\pi(i)})$. Thus, supervision remains aligned with the underlying task: the gradient is simply a convex combination of task-consistent gradients rather than an arbitrary or adversarial signal.
> > >
> > > Second, the injected noise $e_i$ is independent, mean-zero, and isotropic. Because it has no preferred direction and is independent of the labels, it does not introduce systematic bias toward any particular class or feature direction. Instead, it acts as an unbiased perturbation that increases variance but does not distort the supervised objective in a structured way. In this sense, the perturbation behaves more like stochastic regularization or data augmentation than adversarial corruption.
> > >
> > > Intuitively, increasing the noise level makes reconstruction harder, but it does not selectively corrupt task-relevant directions. The supervised gradient signal remains coherent across iterations, and stochastic optimization can average out the isotropic noise. This separation between inversion difficulty and supervised alignment helps explain why accuracy remains stable even when the perturbation magnitude is relatively large.
> > >
> > > We thank the reviewer for pointing this out.

---

### Review · Reviewer_Pjnz · 2026-02-23

**Summary Of Contributions:**

Authors propose a novel algorithm for secure instance encoding for ML training. The algorithm is built upon the previous framework of InstaHide, which is, in turn, built on MixUp where features of multiple input images are overlapped in order to provide protection against reconstruction (and other privacy-oriented) attacks. The framework is evaluated both theoretically and empirically in the context of inversion attacks (2 types) across standard benchmark computer vision datasets. The results demonstrate that the proposed method mitigates the issues identified in prior works in this domain.

**Audience:**

Yes

**Audience Explanation:**

I believe the findings are useful, but (and its a big but) - this would really depend on the application scope of the results. Right now there is a very solid foundation for this - the results of a federated learning setting. Unfortunately, they are not very well-described and neither is the setup of the FL environment. I would have liked to see a lot more, because this is where a lot of value of this work might lie - instance encoding for collaborative learning, of which FL is the most commonly used strategy.

So in principle yes, there is a novel approach to InstaHide which performs better against the selected attacks, but authors did not do their method justice by not discussing the potential implications and limitations in more details. This is something I believe can be addressed in a revision, as if the results are valid it is just the matter of putting the description into the manuscript.

**Broader Impact Concerns:**

In my opinion there is only one potential minor addition, where the authors should state that while under these experimental setups the results reliably show that the shared data is secure, it should not be used in public deployments of sensitive private (e.g. medical imaging) data without proper validation.

**Claims And Evidence:**

Yes

**Claims Explanation:**

I could not identify any obvious issues with the results of this work, so to the best of my knowledge these are valid. I raised some concerns regarding the presentation of results in Requests for changes section, but to summarise: the tables and figures require some additional work with respect to readability and difficulty of interpreting the results (currently the captions are not really helping the reader to understand the figures at all).

**Requested Changes:**

- Please reformulate the abstract, it is currently a lot more about the analysis of other peoples' work rather than about this framework itself. Specifically, please consider removing the references to other authors altogether, just leave the relevant work names (e.g. InstaHide)
- I would like to see a much more detailed section on FL, with setup, more concrete results, specific data splits etc. Currently it reads like an afterthought and to me it is a difference between 'this work is interesting to some/many TMLR readers' based on the scope of the findings of that section. I also am not certain how the state of training (e.g. randomised model vs fully trained model) actually has any effect here. If possible I would like to request a more thorough evaluation of which  specific factors contribute to this framework being more/less suitable in FL (cifar-10 is fine as a choice of dataset), as there are many moving parts and I find the short paragraph stating 'it works' to not be satisfactory.
- Suggestion: you might want to make it super explicit what the noise parameter actually means: in most literature it signifies the strength or the magnitude of the noise, not the threshold, making the figures very difficult to follow (as the intuition is inverted). So either expand the captions or change the way you describe the parameter.
- Figure captions in general require a rework: they really dont help to understand what is going on: which row represents what, etc. Things like 'higher is better' can easily help the reader to understand the context better.
- I would like to ask the authors to check if the reconstruction results hold across random initialisations of the non-linear attack model (i.e. repeat a few times per image). These kind of attacks can be very sensitive to initialisations and I would like to see how well they hold.
- Is there anything to compare the results of this method against InstaHide as a figure? It is not clear to the reader why this method is visually better than the original one (as differences in SNR in pixel space are highly subjective and some variation would not indicate the ineffectiveness of the original method). I would like the authors to show the visual comparison of how their defence performs better.
- Please check for double-referencing (e.g. method proposed by Carlini in Carlini et al), i spotted one in the appendix.

---

> ### Author Response · Authors · 2026-03-03
> **Response - 1**
>
> **Regarding the Abstract**
>
> We have revised the abstract to focus more clearly on our proposed framework and its concrete contributions. In particular, we removed references to specific authors and retained only method names where necessary (e.g., InstaHide), improving clarity and neutrality.
>
> **Regarding Collaborative Training**
>
> We substantially extended the collaborative training section to provide a more systematic and controlled evaluation of the federated setting.
>
> Specifically, we now:
>
> - Clearly describe the full experimental setup: CIFAR-10, frozen pretrained ResNet-18 feature extractor, image-space calibration via Theorem 2, and mapping to a feature-space mixing radius.
> - Explicitly define the collaborative protocol and contrast it with FedProx (50 communication rounds, $\mu = 10^{-3}$, data-size-weighted aggregation).
> - Introduce controlled Dirichlet-based label heterogeneity with $P = 10, 50, 100$ parties and $\beta = 1.0, 0.5, 0.1$.
> - Add a second set of experiments modeling size heterogeneity, where one party is data-poor with size ratio $\rho = 1.0, 0.1, 0.01$ while keeping the total dataset size fixed:
> $$
> n = (P-1)n_{\mathrm{rich}} + n_{\mathrm{poor}} = (P-1+\rho)\,n_{\mathrm{rich}}.
> $$
>
> We now report comprehensive quantitative results under both label skew and joint label+size skew.
>
> **Results under Dirichlet Label Partitioning**
>
> | $P$ | $\beta$ | FedProx | Ours | $\Delta$ |
> |-----|---------|---------|------|----------|
> | 10  | 1.0 | 87.50 | 88.38 | +0.88 |
> | 10  | 0.5 | 87.60 | 88.01 | +0.41 |
> | 10  | 0.1 | 75.10 | 81.14 | +6.04 |
> | 50  | 1.0 | 82.93 | 87.41 | +4.48 |
> | 50  | 0.5 | 83.39 | 84.26 | +0.87 |
> | 50  | 0.1 | 75.21 | 81.73 | +6.52 |
> | 100 | 1.0 | 76.42 | 88.25 | +11.83 |
> | 100 | 0.5 | 76.60 | 88.33 | +11.73 |
> | 100 | 0.1 | 77.35 | 82.18 | +4.83 |
>
> Across all regimes, singularized mixup consistently outperforms FedProx. The improvement generally increases with stronger label skew (smaller $\beta$) and with a larger number of parties.
>
> These results clarify that federated optimization degrades under client drift and heterogeneity, whereas aggregating mixed representations yields a stable centralized training signal without iterative gradient exchange.
>
> **Results under Joint Label and Size Heterogeneity ($P=50$)**
>
> | $\beta$ | $\rho$ | FedProx | Ours | $\Delta$ |
> |----------|----------|----------|--------|----------|
> | 1.0 | 1.0  | 83.09 | 88.27 | +5.18 |
> | 1.0 | 0.1  | 81.56 | 88.67 | +7.11 |
> | 1.0 | 0.01 | 81.27 | 87.22 | +5.95 |
> | 0.5 | 1.0  | 83.54 | 88.04 | +4.50 |
> | 0.5 | 0.1  | 81.51 | 89.05 | +7.54 |
> | 0.5 | 0.01 | 81.70 | 88.56 | +6.86 |
> | 0.1 | 1.0  | 82.07 | 86.22 | +4.15 |
> | 0.1 | 0.1  | 80.55 | 88.36 | +7.81 |
> | 0.1 | 0.01 | 80.23 | 85.08 | +4.85 |
>
> The improvement is particularly pronounced under stronger label skew and moderate size imbalance ($\rho = 0.1$). These experiments broaden the relevance of the work to realistic federated regimes and substantially strengthen the empirical evaluation.
>
> **Regarding the Captions**
>
> We clarified the meaning of the noise parameter in the text, explicitly explaining its relation to the SNR threshold and how it controls noise magnitude.
>
> We also revised and expanded all figure and table captions to clearly describe what each row and column represents and to indicate performance directionality (e.g., “higher is better” where appropriate). This significantly improves readability and interpretability.
>
> **Regarding Initialization for the Nonlinear Attack**
>
> We repeated the nonlinear reconstruction experiments using five different random initializations (random seeds) of the attack model and now report averaged results in Table 3. The conclusions remain unchanged, confirming that reconstruction performance is stable across initializations.
>
> **Regarding the Visual Comparison**
>
> Figures 1 and 2 already provide visual reconstruction results under both linear and nonlinear attacks across different privacy parameters $\tau$, where $\tau$ controls the directional SNR threshold (smaller $\tau$ implies stronger noise and stronger privacy). The reconstructions degrade rapidly as $\tau$ decreases and do not retain meaningful visual fidelity.
>
> Prior work has already established that InstaHide can be inverted to recover images with high visual similarity under their attack model. Since these near-faithful reconstructions are well documented, a direct visual comparison would largely reproduce known results. Our focus is therefore to demonstrate that, under the same attack paradigm, singularized mixup does not allow visually accurate recovery.
>
> **Regarding Double Referencing**
>
> We carefully reviewed the manuscript and corrected instances of double-referencing (e.g., “Carlini in Carlini et al.”), including the case identified in the appendix. References are now consistent throughout.

---

> ### Author Response · Authors · 2026-03-03
> **Response - 2**
>
> **Regarding the limitations of our work**
>
> We acknowledge that the main theoretical limitation of our work is that Theorem 2 provides an *average-case* guarantee. Consequently, there may exist individual samples produced by the singularized mixup (Algorithm 1) that retain more information about the original input than desired. While the theorem ensures that, on average, the directional signal-to-noise ratio is controlled, it does not exclude the possibility of rare outliers.
>
> In practice, particularly in collaborative or distributed settings, this limitation can be mitigated by introducing an additional filtering step after Algorithm 1. Specifically, one can exclude singularized samples that remain too close to their original counterparts according to perceptual or structural similarity metrics. For example, LPIPS, SSIM, or related measures can be used to detect overly similar outputs. In high-stakes applications, such filtering can also be complemented by visual inspection or domain-specific similarity criteria.
>
> To further address this point, we have added a new experiment quantifying the percentage of singularized samples excluded by an LPIPS-based post-processing step (threshold LPIPS < 0.9). For each dataset and privacy parameter $\tau$, we compute the proportion of samples removed due to excessive similarity to the original image.
>
> The results are summarized below:
>
> | **Dataset**   | $\tau=1.0$ | $\tau=0.1$ | $\tau=0.01$ | $\tau=10^{-3}$ | $\tau=10^{-4}$ | $\tau=10^{-5}$ | $\tau=10^{-6}$ |
> |--------------|------------|------------|-------------|----------------|----------------|----------------|----------------|
> | MNIST         | 100.00     | 0.00       | 0.00        | 0.00           | 0.00           | 0.00           | 0.00           |
> | CIFAR-10      | 0.15       | 0.00       | 0.00        | 0.00           | 0.00           | 0.00           | 0.00           |
> | CIFAR-100     | 0.29       | 0.00       | 0.00        | 0.00           | 0.00           | 0.00           | 0.00           |
> | Tiny-ImageNet | 14.06      | 0.00       | 0.00        | 0.00           | 0.00           | 0.00           | 0.00           |
> | CIFAR-5M      | 0.10       | 0.00       | 0.00        | 0.00           | 0.00           | 0.00           | 0.00           |
>
> These results show that for practically relevant security parameters (i.e., $\tau \le 0.1$), the percentage of excluded samples is 0% across all datasets considered. Only for the weakest privacy setting ($\tau=1.0$) do we observe a non-negligible exclusion rate on certain datasets. This empirically supports that, under meaningful privacy configurations, the singularized mixup mechanism does not produce problematic outliers in practice.
>
> ---
>
> **Regarding Deployment and Regulation**
>
> We added the following clarification to the conclusion:
>
> > “As with other data protection mechanisms, practical use must comply with applicable regulatory frameworks such as the General Data Protection Regulation (GDPR) and the Health Insurance Portability and Accountability Act (HIPAA), and should be validated by the responsible legal and compliance entities prior to deployment.”
>
> This explicitly emphasizes that real-world use, particularly in sensitive domains such as medical imaging, requires appropriate regulatory and compliance validation.
>
> We thank the reviewer for the thoughtful and constructive feedback throughout the review process, which has helped us improve the clarity, rigor, and overall quality of the manuscript.

---

### Review · Reviewer_Wf6Q · 2026-02-24

**Summary Of Contributions:**

This paper introduces a method to address the limitation of the defense method InstaHide. Specifically, the method only mixup two private samples and discards the flip procedure in InstalHide, which eliminates the persistent signal that can be exploited by attackers due to the repeated appearance of private samples in multiple mixing processes.
Besides, the paper presents the analysis under the worst-case scenario and the minimax lower bound of the reconstruction error, also shows the selection criteria for the noise norm, providing a theoretical basis for setting noise parameters.
Experiments on MNIST, CIFAR-10, CIFAR-100, and Tiny-ImageNet demonstrate the robustness of the proposed method against attacks.

Strengths
- The paper proposed the defense method, which addresses the limitation of the defense method InstaHide, and also provides a theoretical analysis.
- Experimental results show that the model performance is almost the same as the model without the defense method, while be robust against attacks.

Weaknesses
- Limited validation. It seems that the setup only considers the ResNet architectures. Also, as for the federated learning experiment, what's the data distribution on each client (IID or non-IID)?
- Limited attacks and baselines. It seems that the setup didn't consider the attack scenario where the adversary can modify the network like [3], and compare with other defenses like [1]. It would be better to include more comparisons.
While the proposed method aims to address the limitation in the defense method InstaHide, it would be better to show its effectiveness under broader scenarios.

----
[1] Soteria: Provable defense against privacy leakage in federated learning from representation perspective. CVPR 2021.

[2] Bayesian Framework for Gradient Leakage. ICLR 2022.

[3] Robbing the Fed: Directly Obtaining Private Data in Federated Learning with Modified Models. ICLR 2022.

[4] Inverting gradients how easy is it to break privacy in federated learning? NeurIPS 2020.

**Audience:**

Yes

**Audience Explanation:**

The privacy leakage issue is important, and the proposed method addresses the limitation of the defense method InstaHide, also presents the theoretical analysis, which may bring some insights for other readers.

**Claims And Evidence:**

No

**Claims Explanation:**

The conclusions have rigorous mathematical proofs, and most of the experimental setup is provided.

**Requested Changes:**

- It would be better to include the comparisons with other defenses, as well as other advanced attacks, to verify the effectiveness of the method.
- It would be better to investigate whether the proposed method is still effective when dealing with large-scale datasets.
- As for the federated learning experiment, the details about the data distribution need to be provided.
- It would be interesting if the paper could explore more types of noise.

---

> ### Author Response · Authors · 2026-03-03
> **Response - 1**
>
> **Regarding the Comparison**
>
>
> We clarify that our paper studies privacy-preserving collaborative training via sharing *singularized mixed samples* (or mixed features), rather than iterative federated learning with repeated gradient/model-update exchange. In our protocol, parties do **not** transmit per-round gradients or intermediate model parameters. Instead, each party locally applies singularized mixup and shares only the resulting mixed representations for centralized training.
>
> As a consequence, attack/defense mechanisms designed for gradient-sharing federated learning—e.g., Soteria [1], Bayesian gradient leakage [2], malicious-model attacks such as Robbing the Fed [3], or gradient inversion attacks [4]—operate under a fundamentally different threat surface. These methods assume the server observes client gradients and/or can modify client-side models to extract private information. In contrast, our protocol eliminates iterative gradient exchange entirely, so these attacks are not directly applicable.
>
> For mixing-based schemes, the core security question is whether an adversary can reconstruct original private samples from the disclosed mixed representations. Each released sample corresponds to a noisy linear system in the unknown private data. Therefore, the most relevant and principled attack model is explicit reconstruction by solving this induced linear inverse problem. For this reason, we focus on linear inversion attacks as the primary security evaluation, and we complement them with nonlinear (U-Net–based) learned inversion to account for attackers exploiting data priors. This directly tests whether the injected perturbation renders the inverse problem ill-conditioned in practice, which is precisely the vulnerability exploited in prior attacks on InstaHide.
>
> **Regarding Architectural Choice and Large-Scale Evaluation**
>
> In our experiments, we employ pretrained ResNet backbones (ResNet-18 and ResNet-50) strictly as *frozen feature extractors* to improve computational efficiency and enable extensive experimental sweeps. Backbone weights are never updated, no fine-tuning is performed on either original or mixed data, and the final classification layers are removed. A lightweight classifier is trained on top of extracted representations.
>
> Importantly, the privacy mechanism itself operates entirely in *image space*: noise is calibrated and injected before any feature extraction, according to the theoretical prescription in Theorem 2 (directional SNR bound). The backbone is used solely to reduce training time and computational cost; it is not part of the privacy mechanism and does not influence the theoretical security guarantees. In particular, perturbation is determined from image-space statistics and only mapped to feature space for efficiency, not to enhance accuracy or artificially strengthen security.
>
> To address the concern regarding scale, we extend our security evaluation to CIFAR-5M~\cite{nakkiran2020deep}, which contains approximately 6 million images and represents a substantially larger regime than CIFAR-10/100 or Tiny-ImageNet. On this dataset, we evaluate robustness against the linear inversion attack (the fundamental threat model for mixing-based schemes, since it directly attempts to solve the induced noisy linear system). The results show reconstruction quality remains strongly degraded at this scale, confirming that the injected perturbation continues to render the inverse problem ill-conditioned even as dataset size increases substantially.

---

> ### Author Response · Authors · 2026-03-03
> **Response - 2**
>
> **Regarding Collaborative Training**
>
> We revised the collaborative training section to make the federated setting fully explicit (data distributions, heterogeneity controls, and protocol differences), and we now report comprehensive results.
>
> All collaborative experiments are conducted on CIFAR-10 using the same preprocessing pipeline and a frozen pretrained ResNet-18 feature extractor. The perturbation mechanism is calibrated in image space (Theorem 2) and mapped to a corresponding feature-space mixing radius. Since the backbone remains fixed, differences arise purely from the collaboration protocol.
>
> To control statistical heterogeneity, we adopt symmetric Dirichlet label partitioning. For each class $c$, class proportions across $P$ parties are sampled from
> $$
> \mathrm{Dir}(\beta \mathbf{1}),
> $$
> and examples are allocated accordingly. Smaller $\beta$ induces stronger label skew. We evaluate $P \in \{10,50,100\}$ and $\beta \in \{1.0,0.5,0.1\}$.
>
> | $P$ | $\beta$ | FedProx | Ours | $\Delta$ |
> |-----|----------|---------|------|----------|
> | 10  | 1.0 | 87.50 | 88.38 | +0.88 |
> | 10  | 0.5 | 87.60 | 88.01 | +0.41 |
> | 10  | 0.1 | 75.10 | 81.14 | +6.04 |
> | 50  | 1.0 | 82.93 | 87.41 | +4.48 |
> | 50  | 0.5 | 83.39 | 84.26 | +0.87 |
> | 50  | 0.1 | 75.21 | 81.73 | +6.52 |
> | 100 | 1.0 | 76.42 | 88.25 | +11.83 |
> | 100 | 0.5 | 76.60 | 88.33 | +11.73 |
> | 100 | 0.1 | 77.35 | 82.18 | +4.83 |
>
> Across all regimes, singularized mixup consistently outperforms FedProx, with larger gains as the number of parties increases or label skew becomes stronger.
>
> We further model client size heterogeneity by designating one party as data-poor with
> $$
> n_{\mathrm{poor}} = \rho\, n_{\mathrm{rich}}, \qquad
> n = (P-1+\rho)\,n_{\mathrm{rich}},
> $$
> while keeping the total dataset size fixed. We fix $P=50$ and vary $\beta \in \{1.0,0.5,0.1\}$ and $\rho \in \{1.0,0.1,0.01\}$.
>
> | $\beta$ | $\rho$ | FedProx | Ours | $\Delta$ |
> |---------|--------|---------|------|----------|
> | 1.0 | 1.0  | 83.09 | 88.27 | +5.18 |
> | 1.0 | 0.1  | 81.56 | 88.67 | +7.11 |
> | 1.0 | 0.01 | 81.27 | 87.22 | +5.95 |
> | 0.5 | 1.0  | 83.54 | 88.04 | +4.50 |
> | 0.5 | 0.1  | 81.51 | 89.05 | +7.54 |
> | 0.5 | 0.01 | 81.70 | 88.56 | +6.86 |
> | 0.1 | 1.0  | 82.07 | 86.22 | +4.15 |
> | 0.1 | 0.1  | 80.55 | 88.36 | +7.81 |
> | 0.1 | 0.01 | 80.23 | 85.08 | +4.85 |
>
> In all settings, singularized mixup is more robust to both label and size heterogeneity. Importantly, unlike FedProx, our method is not iterative federated learning: it is a one-shot collaboration where parties share only mixed representations and a centralized classifier is trained once, eliminating repeated gradient/model-update exchange.
>
> **Regarding the Choice of Noise and Comparison to Differential Privacy**
>
> We did not aim to provide a direct comparison with Differential Privacy (DP), as such comparisons were already explored extensively in the original InstaHide work. Our objective here is different: we propose singularized mixup to address a specific vulnerability of mixing-based obfuscation schemes under a realistic and strong threat model.
>
> In particular, we assume the attacker has access to the mixing weights. Under this assumption, each released sample induces a noisy linear system with respect to the original private data, and the adversary can attempt reconstruction by explicitly solving this system. Our noise mechanism is designed to make this inverse problem ill-conditioned and to degrade reconstruction quality even when the mixing structure is fully known.
>
> Unlike DP, which provides formal $(\varepsilon,\delta)$ guarantees via bounded sensitivity and worst-case indistinguishability under neighboring datasets, our approach does not aim to enforce per-sample sensitivity bounds. Instead, the perturbation is calibrated to control directional signal-to-noise ratio and to induce reconstruction instability in the induced linear system. In this sense, our protection notion is reconstruction-oriented rather than sensitivity-based: the goal is to prevent meaningful recovery of private samples while preserving discriminative utility, rather than to satisfy a formal DP guarantee.
>
> We sincerely thank the reviewer for the careful reading and constructive feedback.

---

### Decision · Action_Editor_NqtA · 2026-04-21

**Recommendation:** Accept as is

**Audience:**

Yes

**Audience Explanation:**

This paper touches upon training of neural networks in a privacy-preserving way, which is a main concern of the ML community, and many will welcome reading about these results.

**Claims And Evidence:**

Yes

**Claims Explanation:**

This paper extends the use of mixup training for neural networks for privacy-preserving training. InstaHide, a previous proposal, has some limitations that this paper resolved. Singularized mixup, as this new algorithm is called, mixes exactly two private images while injecting isotropic spherical noise into the non-target component, thereby eliminating the repeated-sample signal that reconstruction attacks on InstaHide exploit. The reviewers presented strong reviews, and most of the comments were addressed by the authors in the rebuttal phase. One reviewer still believes the paper could be improved by conducting additional experiments and robustness tests. However, I believe the paper is comprehensive enough to merit publication as is.